**Data Availability Statement:** All relevant data are within the manuscript and Supporting Information files.

**Funding:** This study was supported by: Funder: Ministero dell'Istruzione, dell'Università e della

# Reflecting glory or deflecting stigma? The interplay between status and social proximity in peer evaluations

Erik Aadland[1], Gino Cattani[2]*, Denise Falchetti[3], Simone Ferriani[4]

1 Department of Strategy and Entrepreneurship, BI Norwegian Business School, Oslo, Norway, 2 Department of Management & Organizations, Stern School of Business—NYU, New York, New York, United States of America, 3 Department of Strategy & Innovation, Questrom School of Business—Boston University, Boston, Massachusetts, United States of America, 4 Department of 'Scienze Aziendali', University of Bologna, Bologna, Italy

* gcattani@stern.nyu.edu

## Abstract

How do candidates' status and social proximity to members of the evaluating audience interact to shape recognition in peer-based evaluative settings? In this study, we shed light on this question by adopting a mixed-method approach. We first examined field data on the conferral of awards in a peer-based evaluative contest–"The Silver Tag"–which is one of the most prestigious digital advertising awards contests in Norway. The field study revealed the existence of a negative interaction between status and social proximity on the allocation of awards. We then conducted two experiments to probe the mechanisms responsible for this finding. In the first experiment, we replicated the main pattern observed in the field study. In the second experiment, we showed that the interaction effect is contingent on the nature of the evaluative setting. When audience members' decisions were in the public domain (i.e., the other audience members knew them), social proximity tempered the effect of status on candidates' recognition, but it did not when decisions were private (i.e., the other audience members did not know them). We conclude by discussing several implications of our study for research on the socio-psychological processes underlying evaluative outcomes in tournament rituals.

## Introduction

Extensive evidence across cultural fields as diverse as academic publishing [1], wine tasting [2], film industry [3], advertising [4], and screenwriting agencies [5] reveals the role of status as a key driver of evaluation and choice. Prevailing explanations for the positive association between status and evaluative outcomes posit that status serves as a source of information about actors' unobserved quality. In this vein, one's relative standing in a social system [6] positively affects others' expectations as well as behaviour toward the object of evaluation. Indeed, high-status actors are assumed to be more competent [7] and more frequently attended to [6]; they are usually granted more recognition for their performance relative to low-status actors

Ricerca Award Number: 2015LJXRXJ Recipient: Simone Ferrriani Funder: H2020 European Research Council Award Number: 695256 Recipient: Simone Ferrriani Funder: Universita' Bologna (IT) Award Number: alma idea Recipient: Simone Ferrriani. No additional external funding was received for this study.

**Competing interests:** I have read the journal's policy and the authors of this manuscript do not have competing interests.

for equivalent performance [8]. By contrast, low-status actors are more likely to be devalued or simply ignored [8, 9]. This explanation has found further support in a few recent studies–more sensitive to the role of the evaluative context–that show how the choice of high-status actors is also more easily defensible before other evaluators because it is based on what is publicly recognized as high quality [10–12].

Social networks are also widely regarded as important drivers of evaluative outcomes. A rich body of empirical research–albeit perhaps not as systematic as the scholarship on status beliefs–supports this view across different evaluative settings in both art and science. Parsons and Shils [13] were among the first to highlight that social relationships between evaluators and candidates may shape reward allocation decisions, and so compromise universalistic standards of evaluation. Blau [14, p. 265] referred to possible social intercourses between evaluators and candidates, emphasizing how "the differentiating criterion is whether the standards that govern people's orientation to each other are dependent on or independent of the particular relationships that exist between them." One of the first studies to find empirical support for this intuition is Wenerås and Wold's [15] analysis of the peer-review system of the Swedish Medical Research Council. They found that postdoctoral fellowship applicants who had relationships with reviewers (e.g., they came from the same academic institution) were judged to be more competent than those who had no such affiliation but were equally productive. Subsequent studies of academic settings have confirmed the existence of a positive association between audience-candidate network proximity and favourable evaluative outcomes [16–18].

Experimental economists have offered complementary evidence in this direction. The work by Dimant [19] on the role of social proximity in magnifying the transfer of norms among peers, and the work by Charness et al. [20] on the role of a salient state of social identity in fostering favouritism towards those of stronger social kinship, are especially relevant. Similarly, scholars in organizational sociology have noted that the attribution of awards to creative professionals in fields of artistic production tends to map on the connectivity between candidates competing for recognition and members of the evaluating audience. Findings supportive of this claim include jurors' preferential allocation of prizes to professionals sharing their networks in the feature film industry [21, 22], and recent findings in the context of the advertising industry exposing the patterning of award allocation choices along relational lines [4].

Overall, there is significant evidence showing how status and social proximity drive audiences' preferential allocation of attention and recognition across competing candidates. Not surprisingly, two of the most widely used truisms to characterize how cultural markets channel resources, honours, and attention to cultural producers reflect precisely the essence of these two mechanisms: "You are as good as your last credit" and "It's Not What You Know. It's Who You Know."

Although we know a great deal about how status and social proximity contribute to producing and reproducing comparative advantages in social evaluations, we know much less about how status and social proximity combine to produce evaluative outcomes. Social ties could either dampen or amplify any positive effect of status on recognition. On the one hand, given the universal nature of status-seeking [23], any positive effect the candidates' status may have on their recognition could be even stronger when high-status candidates have ties to members of the evaluating audience. If recognition flows through the network [18], audience members will extend greater deference to high-status candidates who are connected to them because of the status boost that they themselves can indirectly enjoy via their personal affiliation to those actors. This tendency to strive for prestigious affiliations has been described as the "basking-in-reflected-glory effect" [24], or "basking in the reflection of a neighbor's glory" [25], and it is based on the way individuals extend their perceptions from one subject to another. This logic implies that, all else being equal, the marginal effect of candidates' status on their recognition

will increase monotonically with their social proximity to audience members. On the other hand, the existence of social ties between audience members and candidates could reduce the saliency of status as a signal of the quality of candidates and their work. Status signals are most valuable when there are no or limited alternative ways of evaluating a firm's actual quality [6]. So, when a candidate projects information both due to her status and her direct connection to the audience, one of those information devices may be redundant, as market participants are likely to satisfice on collecting and processing information [26]. Consistent with this idea, prior evidence at the organizational level of analysis has established that relationships that firms maintain with their customers and status are *substitutive* drivers of market entry decisions. Both types of social resources facilitate entry into a new market, but the importance of status diminishes in the presence of market ties, which "represent a more direct mechanism than [. . .] status to reduce market uncertainty and increase exchange value" [27, p. 467]. Hence, assuming that status serves as a quality signal, we should expect social proximity to reduce the marginal effect of status on recognition.

Recent findings by Aadland, Cattani, and Ferriani [4] imply similar expectations but rely on different interpretations. While providing substantial evidence supporting the general stratifying effect of audience-candidate ties on candidates' recognition, the authors also point to the plausibility of negative returns of social proximity to recognition–particularly at high levels of audience-candidate social proximity. The authors attribute this possibility to what they call "intellectual distance," a term used to indicate audience members' attempt to project their "interest in disinterestedness" [28, p. 112, see also pp. 87–88]. The general intuition is that the audience-candidate proximity in the social network might give rise to morally problematic interpretations of audience members' true intentions and yield reputational concerns that inhibit favourable evaluations of socially proximate candidates. Following this logic, it is then plausible to expect audience members' reliance on status cues to decline as social proximity increases. If intellectual distance kicks in, in fact, the glory enjoyed by the target of deference will deflect stigma on the evaluator. Thus, audience members might feel that at higher levels of social proximity elevating the status of candidates could more easily backlash. Even these arguments suggest a negative interaction effect between status and social ties, but rather than being the result of substitutive dynamics, the effect here derives from self-motivated concerns. In summary, how the status of candidates and their social proximity to audience members interact in evaluative contexts is unclear due to the coexistence of alternative perspectives that indicate different explanatory mechanisms. Which of these perspectives best characterizes the interaction effect between status and social proximity on the evaluative outcomes?

Our objective in this paper is to shed light on this theoretical question. To do so, we collected data on the conferral of prestigious awards to competing candidates in a peer-based evaluative contest–"The Silver Tag"–which is one of the most prestigious digital advertising awards contests in Norway, and examined how candidates' status and audience-candidate proximity in the underlying social network, and their interaction, explain award allocation decisions. In this field study, we find a negative interaction effect between candidates' status and audience-candidate proximity on the conferral of awards. We offer external validity to the field study and evidence on the mechanism responsible for the field study's results by supplementing them with two experiments. The first experiment replicates the main pattern of the effects observed in the field study, which indicates a negative interaction between status and social proximity. The second experiment reveals how the interaction effect is contingent on the nature of the evaluative setting. In doing so, we seek to distinguish–theoretically and empirically–processes associated with stigmatic perceptions from alternative explanations that imply the same empirical patterns but rely on different assumptions about the interplay between status and social ties. We show that when the evaluation is public–and so potential

violations of the meritocratic ideal in social evaluation are easier to detect and stigmatize, if not punish–social proximity mitigates the effect of status on candidates' recognition, but it does not when those decisions are private (hence audience members do not have to discloser and justify their decisions before the other members). We conclude by discussing the implications of this study for research on the socio-cognitive processes underlying the evaluation of peers in ostensibly meritocratic settings and by identifying avenues for future research.

## Overview of studies

To examine the interplay between status and social ties in peer-based evaluative settings, and the socio-cognitive drivers of recognition, we used a mixed-method approach. Specifically, we conducted one field study (Study 1) and two experiments (Study 2 and Study 3) to ensure the internal and external validity of our studies. We conducted the field study in a setting where status and social ties shape peer audience evaluations, but neither the status of the candidates nor their social ties to the members of the evaluating audience were manipulated. This feature of the field study allows us to establish whether status and social ties are additive (reinforce each other's effect) or substitutive (reduce the effect of one when in the presence of the other).

The two experiments further examine the interaction effect and the conditions under which this effect is more or less likely observed. For the experimental studies, approval by the ethics committee was not required because the data were analyzed anonymously and we did not collect any personal identifiable information. Each participant also filled out an online consent form and voluntarily agreed to participate. Study 2 explores the joint effect of status and social ties on the probability of rewarding cultural works by manipulating the status of the candidates and the social ties between the candidates and the members of the evaluating audience. Study 2 replicates the findings of the field study, though with stricter quality controls. (One crucial strength of the experimental approach is the possibility of holding the project quality constant. The true quality of cultural producers' work is unobservable and difficult to infer unequivocally even after consumption. The challenge is to adopt an approach that allows the researcher to ascertain the presence of evaluative drivers independent of the quality of the producers' work. Study 3 probes the interaction effect by holding the presence of social ties between candidates and audience members constant, and manipulating the candidates' status and the transparency of the evaluation process. Study 3 sheds light on the circumstances under which the negative interaction is more or less likely to operate, offering precious insights into the nature of the mechanism underlying what we observed in the field data.

## Study 1

We examined the interplay between status and social ties in peer audience evaluations. As the advertising industry is project-based it is not uncommon for jury members to evaluate peers with whom they collaborated in the past.) in the Norwegian advertising industry, where it is customary to establish excellence through awards contests [29, 30].

### Interviews with key informants

We interviewed a panel of field insiders consisting of élite advertising professionals, advertising professionals struggling to make their mark, advertising awards contest jurors, and representatives from industry associations (We gained access by first presenting our project to the main advertising organizations in Norway: Kreativt Forum and INMA. After securing their support, we approached a sample of agencies that varied on key dimensions of interest for our study and asked if they were willing to participate. All agencies agreed to participate and gave us access to key advertising professionals. Each interview lasted between twenty minutes and

**Table 1. Descriptive data on agencies sampled for interviews.**

| Agency | Agency size | Services | Digital awards | Respondent's role |
|---|---|---|---|---|
| Advertising | 75–100 | Full service[a] | Yes | Digital advisor |
| Advertising | 75–100 | Full service | Yes | Copywriter; Managing director |
| Advertising | 50–75 | Full service | Yes | Digital advisor |
| Advertising | 40–50 | Mass communication | Yes | Copywriter |
| Advertising | 10–20 | Mass communication | No | Copywriter |
| Advertising | 10–20 | All media | No | Art director |
| Digital | 20–30 | Digital | Yes | Art director |
| Digital | 5–10 | Digital | No | Digital advisor |
| Digital | 5–10 | Digital | Yes | Digital advisor |

[a] Full-service agencies typically offer a wide range of services such as mass communication, direct marketing, digital, design, and sometimes media brokering.

two hours. The topics covered during the interviews included collaborative practices, meaning and relevance of awards, advertising evaluation criteria, perceptions of distance or proximity in the social space, personal anecdotal evidence of jury decisions, and deliberation processesWe interviewed some of the interviewees several times to further probe their field experience. In total, we conducted 19 interviews and followed up with interviewees by email to validate our interpretation of the data. None of the organizations, agencies, or professionals received any compensation. Although these interviews did not form a representative sample of industry participants' opinions, a considerable range of views was expressed and noteworthy themes emerged that enhanced our understanding of the award contests' evaluative dynamics. Table 1 reports descriptive data on the sampled agencies and respondents. Our industry informants suggested that professionals' social standing in the professional status hierarchy is a signal of their (uncertain) quality. This status information, in turn, influences jury evaluations as the following quote by a copywriter and former juror in an advertising agency illustrates:

> "It's a bit like that [well known high status creative teams] have a tendency to score incredibly well on work that is really only average. And that is because you are positively biased, because they make a lot of nice work. And you are a bit positively biased to begin with. You really want the work they do to be of high quality. And, sure, if you come in [to an awards contest], if you send in something from [an out of town agency] that is not highly regarded in the industry, then you will struggle a lot."

Our informants also knew that jury deliberations often are enveloped into "interpersonal patterns of value commitments" that channel attention, energy, and information, while subtly shaping attributions of ability [4, p. 893]. The following quote [4, p. 893] from an account manager is telling:

> "If two projects are equally good, then the project where project members and jurors know each other will win [. . .] these people share the same opinion about what is "important" and "not important," as well as what is "right" and not "right." They [the projects by candidates previously tied to the jurors] might, therefore, score higher on the criteria valued by the jurors who 'administer the truth' about what is good and not so good."

Our informants recognized the influence of professionals' status and ties to jury members in shaping such jury's evaluations of their work. However, they were also deeply aware that the identity of the jury members is public information available to colleagues in the industry and that the professional relationships between the jury members and the candidate producers are relatively transparent with the other members of the industry, in particular other jury members take part in award decisions. In this type of socio-relational context, the social ties of a jury member with a candidate can sometimes translate into more of a liability than an advantage. Our informants have stressed that susceptibility to claims against impartiality in evaluations also tends to influence the results of the jury's deliberations. An experienced jury member suggested how voicing a genuine preference for a particular project could potentially become a source of stigma due to prior collaborations with some project-team members [4, p. 894]:

> "It is a big problem if they [i.e., the members of the industry] come to believe you have a vested interest. If you favour that project [. . .] you may end up in big trouble. I usually keep quiet or alternatively try to mention what is good about other projects in such situations."

In summary, our interviews seem to reveal a fundamental evaluative ambivalence caused by the strong susceptibility of jurors to claims against their authenticity. Avoiding conflicts of interest can be a matter of moral conviction or adherence to epistemic values. The composition of the jury is in the public domain; likewise, the existence of professional relationships between jury members and candidate producers is relatively visible to other members of the industry as professionals have a good sense of who has worked with whom. Lurking suspicions of deliberations along these relational lines, therefore, can easily emerge and question their moral character, even when the members of the jury sincerely approve these deliberations.

## Secondary data

We investigate the interaction effect between status and social ties in peer audience evaluations using the novel "The Silver Tag" dataset first described in Aadland [31]. The dataset includes all projects entered into "The Silver Tag"–the monthly Norwegian digital advertising awards contest–from May 2003 to April 2010. The data comprise a total of 1,734 distinct individuals, 350 distinct organizations, and 902 projects corresponding to 11 competitions per year and 75 contest months over the study period. The Norwegian interactive marketing interest organization responsible for the contest, INMA, combines the contest months June and July each year into one contest generation. Also, INMA combined March/April 2004 and August/September 2004 into two distinct contest generations. The data contain all winners, recipients of honorable mentions, and losers. The data also track all jury members serving on juries in "The Silver Tag" awards contest from May 2003 to March 2010. Each jury served from May to April of the following year during the years 2003–2006 and from April to March during the years 2006–2010. In total, the dataset contains 7 juries, whose size over the study period varied from 4 (for the first jury) to 11 (for the last jury) members.

## Dependent variable

Following Aadland et al. [4], the dependent variable measures the bestowal of an accolade (honourable mention or award) to projects competing in a given contest month. We coded the dependent variable 0 if a project did not receive any accolade; 1 if a project received an honourable mention; 2 if a project reached 1st place (i.e., won the award). The dependent variable is ordered in terms of levels, or intensity, of peer recognition.

## Independent variables

**Social ties.**   We captured the effect of social proximity between audience members and candidates on the likelihood of receiving an accolade by looking at the impact of direct ties. We observed direct ties when the project and jury members had worked on the same project(s) in the past. We calculated this variable by first generating bipartite project affiliation network matrices based on the monthly digital awards contest "The Silver Tag" using Ucinet, version 6 for Windows [32]. We created the adjacency matrices with a 24-month long moving window that we updated monthly. Because our unit of analysis is the project, for each project we created the variable *social ties* by counting only the number of jurors with direct ties to project members [4]. We also looked at the impact of having mediated (i.e., indirect) ties to jury members on the likelihood of being rewarded by calculating the median geodesic distance between the project and jury members. Following Aadland et al. [33, p. 140], we first calculated the median geodesic distance between each project member and the jury members. Consistently with the six degrees of separation theory [34], we then grouped individual producers with a degree of separation from jurors equal to or greater than 6, and assigned them the value 6. To facilitate the interpretation of the results, we measured the variable in terms of nearness between jury members and producers by calculating the reciprocal of the median geodesic distance between each project member and the jury members. As our unit of analysis is the project, we created the *social proximity* variable by taking the median of each project member's median distance from jury members.

**Project status.**   We relied on network centrality to measure status in line with previous research (for a review see [35]). We created the project status variable using Bonacich beta-centrality [36], a measure that is commonly used to derive status ordering from relational data [37–40]. The beta-centrality measure captures a professional's prominence within the peer network as a function of both the number and the centrality of professional peers to whom s/he is connected. The status of these peers is, in turn, a function of the number and centrality of the professional peers connected to them, and so on. Hence, the beta-centrality scores determine each professional's position within the global network's status hierarchy. When beta is set to zero, network centrality is akin to degree centrality, focusing only on the local structure. The larger the value of beta, the more the centrality measure reflects the global structure. In our analysis, we set beta to the reciprocal of the largest eigenvalue. However, even considering a range of values for b, we found no substantial differences in the status scores. We used UCI-NET version 6 [32] to calculate our status measure. Our *project status* measure counts the number of professionals in the project with a Bonacich beta-centrality above the median in the global "The Silver Tag" network over the total number of individuals working on the same project in a particular contest month. We calculated our centrality scores based on the same 24-month long moving affiliation network window we used for the social ties measures. We also chose a more conservative cutoff to define high-status–i.e., values greater than .85 (for a similar approach see [41])–which yielded very similar results.

## Control variables

To rule out alternative explanations for the hypothesized relationships, we included several control variables in our models.

**Project sophistication.**   In "The Silver Tag," jury members typically emphasize whether the advertising projects competing in a given contest month use new technology. The creative use of technology is perceived as a sign of technical sophistication and innovativeness that, in turn, represents a sign of higher quality digital advertising projects. Accordingly, the variable *project sophistication* differentiates projects based on the type of technologies that they

employed. Following Aadland et al. [33, p. 141], "the variable counts the number of agencies specializing in 3D-animation, film production, radio production, or back-end streaming involved in a given project." While not capturing directly the use of new technologies, this variable identifies projects for which those technologies in principle could have (and most likely were) employed.

**Project size.** We controlled for the total number of individuals on each digital advertising project because the number of project participants serves as a proxy for larger project budgets and more available resources to create projects of higher quality.

**Conflict of interest.** Jury members are not allowed to partake in the evaluation of a project whenever they have a conflict of interest in that project. For instance, when the project and jury members work for the same firm or jury members are involved in projects under evaluation, the jury member in question has to exit the jury room when the project in question is evaluated. Accordingly, we generated an indicator variable that is equal to 1 if one or more project members had a colleague in the jury or a juror was a member of the project, and 0 otherwise [4, 33].

**Prior positive co-experience.** Some jurors may have collaborated with candidates and won with them on projects in the past. If prior candidate-juror interactions have resulted in the achievement of a positive outcome they are likely to affect the evaluators' disposition toward the work of their past collaborators when the jurors in question cast their vote over the competing candidates [4]. Previous social network research has shown how social ties can be a source of social benefits (e.g., more favorable evaluations) or social liabilities (e.g., less favorable evaluations) depending on whether relationships between evaluators and candidates are positive or negative [42]. We then identified "The Silver Tag" projects in which a current candidate and a juror collaborated and won the award during the prior 24 months. We created the indicator variable *prior positive co-experience*, which is equal to 1 if there were one or more such instances for a given project, and 0 if there were no such instances.

**Project median experience.** Project members' past experience with digital advertising projects might account for their differential ability to contribute to the project and understand what exactly jury members are looking for in a project. We measured project members' past experience by tallying the number of projects before the focal project each producer had submitted to "The Silver Tag" contest. For each project, we then calculated the *median experience* of all producers involved [4, 33].

**Competitive intensity.** The more projects compete for recognition in a given contest month, the more intense the competition and the lower the likelihood that a given project will win [4]. We controlled for *competitive intensity* by counting the number of projects competing in each contest month.

**Reciprocity.** Reciprocity, the giving of gifts to another in return for gifts received, is also a distance-reducing mechanism between any two parties involved in a social exchange [43]. As Sherry [44, p. 158] observed, "The giving of gifts can be used to shape and reflect social integration (i.e., membership in a group) or social distance (i.e., relative intimacy of relationships)." Accordingly, we created the *reciprocity* variable that "captures the extent to which jury members reward projects whose members were jurors in the past and who–in that role–had rewarded one or more of the current jury members" [4, p. 897]. For each project, the measure counts the number of current jurors who won or received an honorable mention by project members serving as jurors over the previous two years and whose work happened to be under evaluation during the focal contest month.

**Jury status.** We measured the status of the jury by counting the number of jurors in the jury with a Bonacich beta-centrality above the median in the global "The Silver Tag" network over the total number of jury members in a particular contest month.

**Jury median experience.** We measured jurors' past experience by counting the number of projects each juror had submitted to "The Silver Tag" contest before the focal month. For each jury and contest month, we then calculated the *median experience* of all jurors involved.

## Method

We modelled the probability of each project receiving more favourable evaluations by the jury members in a given contest month by using generalized linear models [45, 46]. We estimated our models with the glm command in Stata 14, specifying the binomial family and setting the binomial denominator equal to the number of jurors evaluating the competing projects in each contest month [33]. We also specified the logit link and estimated our models with maximum likelihood. We clustered the standard errors on contest month to obtain robust standard errors. For each contest month, we modelled the probability of jury members assigning an outcome for each project of either no placement (0 points), honourable mention (1 point), or winning the award (2 points).

## Results and discussion

We report descriptive statistics and correlation values for our measures in Tables 2 and 3, respectively. Since the condition number [47] for the matrix of independent variables to be 10.04 –well below the suggested threshold of 30 –multicollinearity is not likely to be an issue in our models.

We began by estimating a model with robust standard errors in which the only predictor is *project status*. The model stratifies by contest month, so each stratum corresponds to a choice set for the jury in a particular month. In Model 1 of Table 4, the coefficient for *project status* is 1.062 ($p<.01$). We then estimated a model with *social ties* only. In Model 2, the coefficient for *social ties* is .322 ($p<.01$). The pattern and significance of the two predictors remain stable when both variables are included together (Model 3). We then proceeded to estimate the interaction between *project status* and *social ties* and the main effects for the interaction term components. In Model 4, the coefficient for the *project status* x *social ties* interaction is -.420 ($p<.01$), while the coefficient for *status* is 1.189 ($p<.01$) and the coefficient for *social ties* is .503 ($p<.01$). The negative joint effect of status and ties suggests that jury members are less inclined to reward high-status candidates who are socially close to them.

**Table 2. Descriptive statistics.**

| Variables | Mean | Std. Dev. | Min | Max |
|---|---|---|---|---|
| 1. Allocation of rewards | .408 | .631 | 0 | 2 |
| 2. Project size | 7.034 | 3.818 | 1 | 30 |
| 3. Project sophistication | .225 | .559 | 0 | 5 |
| 4. Median experience project | 3.751 | 4.149 | 0 | 30 |
| 5. Competitive intensity | 15.364 | 5.968 | 3 | 30 |
| 6. Conflict of interest | .436 | .496 | 0 | 1 |
| 7. Prior positive co-experience | .261 | .440 | 0 | 1 |
| 8. Reciprocity | .281 | 1.208 | 0 | 7 |
| 9. Median experience jury | 5.706 | 3.061 | .5 | 12 |
| 10. Jury status | .439 | .126 | .1 | .7 |
| 11. Project status | .441 | .402 | 0 | 1 |
| 12. Social ties | 1.024 | 1.179 | 0 | 5 |

**Table 3. Correlation coefficients.**

| Variables | 1 | 2 | 3 | 4 | 5 | 6 | 7 | 8 | 9 | 10 | 11 | 12 |
|---|---|---|---|---|---|---|---|---|---|---|---|---|
| 1. Allocation of rewards | 1 | | | | | | | | | | | |
| 2. Project size | .32 | 1 | | | | | | | | | | |
| 3. Project sophistication | .25 | .38 | 1 | | | | | | | | | |
| 4. Median experience project | .12 | -.07 | -.04 | 1 | | | | | | | | |
| 5. Competitive intensity | -.21 | -.06 | -.01 | -.00 | 1 | | | | | | | |
| 6. Conflict of interest | .19 | .23 | -.03 | .22 | -.09 | 1 | | | | | | |
| 7. Prior positive co-experience | .19 | .27 | .16 | .25 | -.03 | .55 | 1 | | | | | |
| 8. Reciprocity | .34 | .18 | .06 | .18 | -.15 | .21 | .22 | 1 | | | | |
| 9. Median experience jury | .14 | .21 | .09 | .14 | -.08 | .04 | .10 | .19 | 1 | | | |
| 10. Jury status | -.00 | .04 | .09 | -.04 | .18 | -.04 | .01 | .06 | .27 | 1 | | |
| 11. Project status | .27 | .35 | .17 | .29 | -.05 | .28 | .38 | .17 | .03 | .01 | 1 | |
| 12. Social ties | .28 | .35 | .15 | .31 | -.07 | .59 | .71 | .23 | .17 | .03 | .50 | 1 |

Condition number = 10.04

The next model includes our control variables (Model 5). While *project size*, *project sophistication*, *competitive intensity*, and *reciprocity* are significant and the sign of their coefficient in the expected direction, the other controls are not statistically significant. When all these variables are controlled for (Model 6), the coefficients for *project status* and *social ties* are positive and significant. Model 7 presents the results of the full model including the controls, the interaction components, and the interaction effect. The coefficient for the interaction term is -.471 ($p<.01$), corresponding to an odds ratio of .624 and a 37.6% decrease in the odds of gaining recognition. The main effect coefficient for *project status* is 1.025 ($p<.01$), corresponding to an odds ratio of 2.787, and the coefficient for the main effect of *social ties* is .485 ($p<.01$) which corresponds to an odds ratio of 1.623. We also calculated the marginal effect of *social ties* for representative values of *project status* to further explore their interplay. Fig 1 plots this marginal effect. The plot reveals a positive marginal effect of *social ties* that decreases for higher levels of *project status* and eventually turns insignificant for very high levels of *project status*. Similarly, Fig 2 plots the average marginal effect of *project status* at representative values of *social ties*. As the proportion of project members with direct ties to members of the jury increases (i.e., the value of the variable gets closer to 5), the marginal effect of *project status* on receiving an honorable mention or winning (i.e., outcomes 1 and 2) decreases, suggesting that direct ties to jury members become increasingly important in shaping their reward allocation decisions. For values of *social ties* equal to or greater than 2, the marginal effect of *project status* is not significant.

We also calculated the adjusted predictions for the number of *social ties* at representative values of *project status*, holding the other variables constant at their means. Fig 3 plots the adjusted predicted probabilities. When *social ties* = 5 and *project status* = 0, the adjusted predictive margin is 1.639 ($p<.01$). Conversely, when *social ties* = 5 and *project status* = 1, the adjusted predictive margin is .493 ($p<.01$). The adjusted predicted probabilities suggest that the likelihood of reward is high for projects with higher levels of *social ties* and low levels of *project status*. The likelihood of rewarding projects with higher levels of *social ties* decreases when *project status* increases. By contrast, when *social ties* = 0 and *project status* = 0, the adjusted predictive margin is .171 ($p<.01$). When *social ties* = 0 and *project status* = 1, the adjusted predictive margin is .462 ($p<.01$). The adjusted predicted probabilities suggest that

**Table 4. Generalized linear models (clustered on contest/month).**

| Variables | Model 1 Coeff. | Model 2 Coeff. | Model 3 Coeff. | Model 4 Coeff. | Model 5 Coeff. | Model 6 Coeff. | Model 7 Coeff. | Model 8 Coeff. |
|---|---|---|---|---|---|---|---|---|
| Project size | | | | | .066*** | .047** | .045** | .047** |
| | | | | | (.015) | (.017) | (.016) | (.015) |
| Project sophistication | | | | | .317*** | .308*** | .337*** | .265*** |
| | | | | | (.069) | (.078) | (.079) | (.062) |
| Median experience project | | | | | .028+ | .007 | .014 | .013 |
| | | | | | (.015) | (.016) | (.015) | (.015) |
| Competitive intensity | | | | | -.047*** | -.045*** | -.044*** | -.049*** |
| | | | | | (.011) | (.010) | (.009) | (.010) |
| Conflict of interest | | | | | .245+ | .127 | .049 | .025 |
| | | | | | (.142) | (.147) | (.141) | (.141) |
| Prior positive co-experience | | | | | -.038 | -.345+ | -.300+ | -.127 |
| | | | | | (.157) | (.181) | (.170) | (.152) |
| Reciprocity | | | | | .133*** | .134*** | .139*** | .130*** |
| | | | | | (.028) | (.027) | (.027) | (.025) |
| Median experience jury | | | | | .005 | .013 | .009 | .002 |
| | | | | | (.014) | (.014) | (.014) | (.016) |
| Jury status | | | | | -.131 | -.237 | -.127 | .004 |
| | | | | | (.363) | (.333) | (.310) | (.338) |
| Project status | 1.062*** | | .740*** | 1.189*** | | .546** | 1.025*** | 3.115*** |
| | (.145) | | (.182) | (.250) | | (.202) | (.260) | (.755) |
| Social ties | | .322*** | .208*** | .503*** | | .154* | .485*** | |
| | | (.038) | (.050) | (.089) | | (.069) | (.104) | |
| Project status * Social ties | | | | -.420** | | | -.471*** | |
| | | | | (.133) | | | (.126) | |
| Social proximity | | | | | | | | 9.408*** |
| | | | | | | | | (1.595) |
| Project status * Social proximity | | | | | | | | -10.245*** |
| | | | | | | | | (2.505) |
| Constant | -3.727*** | -3.582*** | -3.824*** | -4.066*** | -3.385*** | -3.506*** | -3.789*** | -5.639*** |
| | (.116) | (.074) | (.109) | (.151) | (.267) | (.260) | (.270) | (.480) |
| N | 654 | 654 | 654 | 654 | 654 | 654 | 654 | 654 |
| Log pseudolikelihood | -516.94 | -517.81 | -509.68 | -505.09 | -481.584 | -474.14 | -469.16 | -463.52 |
| AIC | 1037.87 | 1039.61 | 1025.37 | 1018.18 | 983.15 | 972.29 | 964.31 | 953.05 |

Robust standard errors in parentheses

*** p<0.001,

** p<0.01,

* p<0.05,

+ p<0.10

the likelihood of being rewarded is low for projects with lower levels of *social ties* and low levels of *project status*, but that the likelihood increases slightly when *project status* increases.

Overall, these results identify an important boundary condition that may alter the saliency of status cues. While social ties appear to reduce the need to rely on status-based evaluation, as encoded in a publicly observable status hierarchy, they become less salient in driving recognition as the project members' status increases. These results, in other words, suggest that status

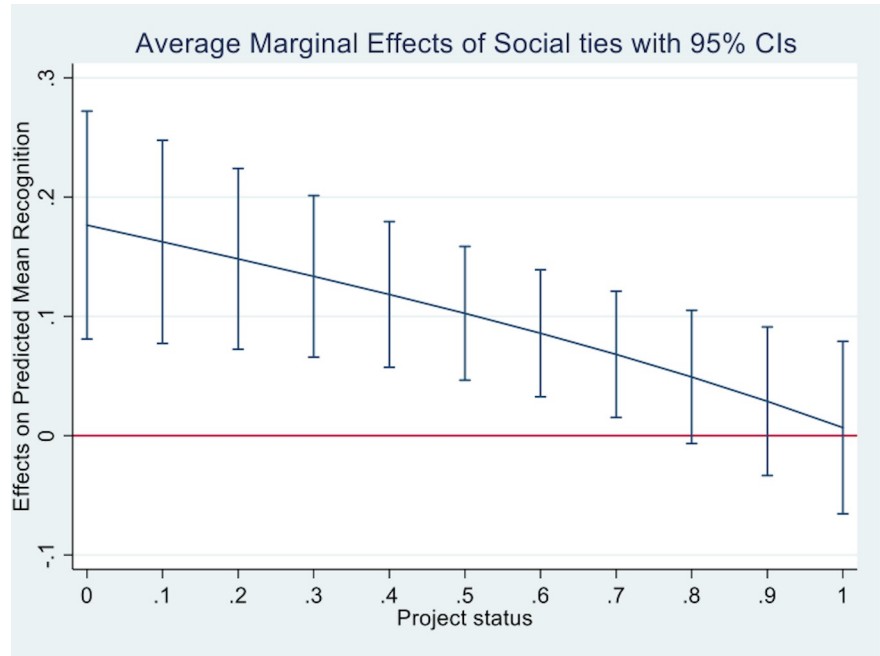

**Fig 1. Average marginal effects of *social ties*.**

in the project and social ties are not additive, pointing instead to a substitution effect between them.

**Robustness checks.** We conducted additional analyses to gauge the validity of our findings. In particular, we looked at the impact of having mediated (i.e., indirect) ties to jury

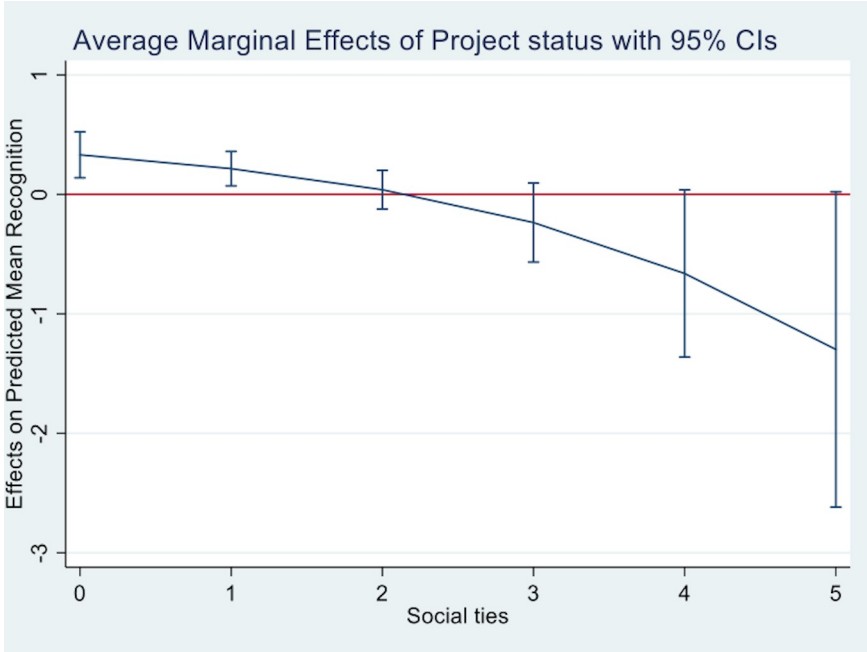

**Fig 2. Average marginal effects of *project status*.**

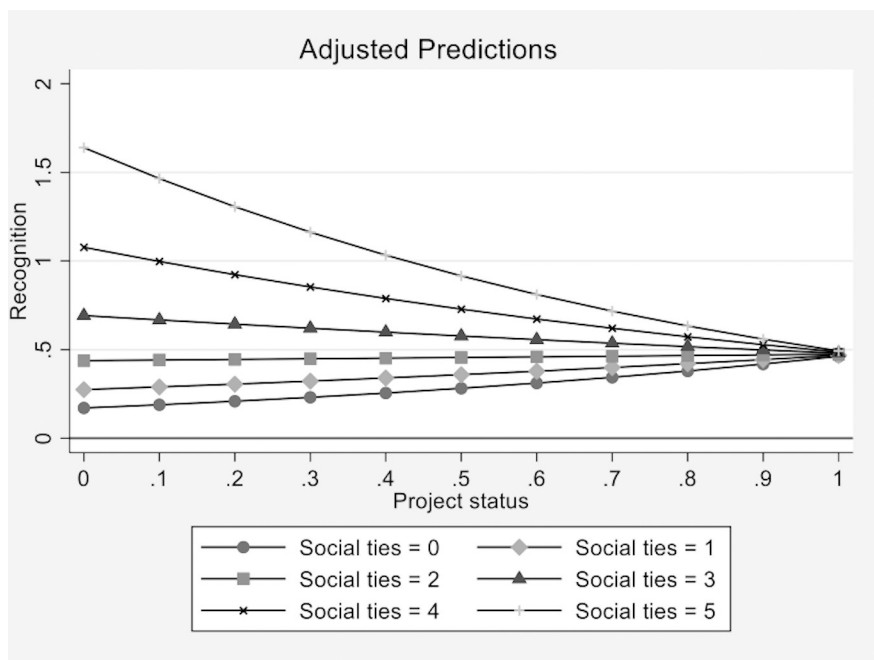

**Fig 3. Adjusted predictions for the number of *social ties* at representative values of *project status*.**

members on the likelihood of being rewarded. We re-estimated the full model (Model 8) by interacting the alternative measure of *social proximity* with the *project status* variable. The pattern and significance levels in the model remain stable: the coefficient for the interaction term is -10.245 ($p<.01$), while the main effect coefficient for *project status* is 3.115 ($p<.01$) and the main effect coefficient for *social proximity* is 9.408 ($p<.01$). These results also confirm the existence of a substitution effect between status and social (direct and indirect) ties. We also re-estimated the full model (Model 8) after orthogonalizing all explanatory variables and the results were the same.

To provide an alternative control for the effect of project quality, we estimated the full model 7 with a latent *project quality* variable. The construction of the latent project quality variable is described in detail in [4]. The coefficient for *project quality* is positive, 1.336, and significant at ($p<.01$), while the overall pattern and significance levels remain stable. The coefficient for the interaction term is -.423 ($p<.01$), the main effect coefficient for *project status* is .902 ($p<.01$) and the main effect coefficient for *social ties* is .449 ($p<.01$).

We also collapsed the levels of the dependent variable–with non-wins = 0 and recognition (honorable mention or win) = 1 –and re-estimated the full model 7 with fixed effects conditional logistic regression. The variables *competitive intensity*, *median jury experience*, and *jury status* are dropped from the model due to lack of within contest month variance, but the pattern and significance levels for the remaining variables in this model also remain stable. Exposing the interaction effect, the coefficient for the interaction term is -.546 ($p<.05$), while the main effect coefficient for *project status* is 1.081 ($p<.01$), and the main effect coefficient for *social ties* is .632 ($p<.01$). Finally, the results were not affected when we clustered the standard errors on firm rather than contest/month. Although the results for the last analysis are not reported here, they are available from the authors upon request.

## Study 2

During our observation window, the advertising field insiders we interviewed emphasized how projects of high quality were likely to exhibit certain measurable attributes separate from the un-measurable idiosyncratic aspects of the creative idea underlying each project. Accordingly, in the field study, we controlled for some of these project-level attributes. Yet, other unobserved characteristics not captured in our analysis might affect jury members' perception of project quality, thereby affecting the chances of a project being rewarded. In Study 2, we tried to alleviate this concern by replicating our effects in an online experimental study. By asking participants to evaluate the same advertising project–thus keeping its quality constant and varying only descriptions of candidates' status and their social ties to evaluators–Study 2 helps rule out quality differences among projects as an explanation for our results. We replicated the field study by priming all participants to think that their evaluations were in the public domain and telling them that the jury would select the winners collectively. We developed vignettes to describe an award contest–i.e., a fictional Digital advertising contest–in which we asked participants to serve as jury members and bestow an award on a commercial. In the vignettes, we employed different cues to manipulate the status level (status vs. no-status) of the commercials' creators, and the presence of social ties (direct ties vs. no-direct ties) between them and the experiment participants (i.e., jury members). We used *award propensity* as the dependent variable.

### Method

**Participants.** Six hundred and fourteen participants were recruited online using Amazon's Mechanical Turk. They received $1.00 for completing the study. Potential participants were restricted only to US residents with a 95% or greater approval rating on MTurk. To ensure that participants read and completed the questionnaire carefully, we applied two attention checks and excluded from the final analysis participants who missed the correct answers. To check if participants really watched the commercial, we asked them "What is the commercial about?" after they watched the video. Specifically, they had five options as possible answers: Financial Service (the correct answer), Nutrition Service, Medical Service, Recycling Service or Health Care Service. The second check was an instructional manipulation check [IMC, 48] to ensure participants read the text: specifically, we instructed participants to leave blank the following question: "Do you think commercials affect your purchasing decisions? Please justify your answer with an example." Since we required participants to watch a commercial that lasted 55 seconds, we removed those participants who did not watch and/or spent too much time watching the video. We then recorded the time each participant spent on the page with the commercial and computed the percentiles for the time variable. In the analysis, we used the data on the participants included in the 5th and the 95th percentile–which corresponded to 50 and 122 seconds, respectively. All these procedures are strongly recommended to ensure the pool of subjects is of high quality and remove inattentive responses when online tools such as Mechanical Turk are used [49–53]. The final sample consisted of 518 participants (52.5% female, $M_{age}$ = 36.53 years, 75.3% Caucasian).

**Material and procedure.** We randomly assigned participants to one of the four conditions in a 2 (status: status vs. no-status) x 2 (social ties: direct ties vs. no-direct ties) between-subjects experiment. Participants first read a vignette that informed them about a digital advertising competition where they had to serve as jury members. Then, they were asked to assign an award to a commercial after evaluating its aesthetic beauty and animation features. We chose these two evaluative criteria because they represent the qualities evaluated by the jurors in our field study. In order to replicate the evaluative process of the field study, we also

informed the participants that "the jury selects the winner collectively thereby disclosing the vote cast by each jury member." The purpose of this clarification is to induce the participants to think that they have to justify their evaluation before the other jury members. We used this vignette to describe the evaluative setting:

### Advertising digital competition

"In your community, there are many initiatives, including an annual Competition in Digital Advertising. Everyone in the community can participate in the competition by submitting a commercial. Each commercial is judged and has the opportunity to win an award. Since you participated in the competition in the past, this year the organizers of the competition have asked you to become a **jury member**. As a jury member, you have to assign an award to a commercial after evaluating its **aesthetic beauty** and **animation features**."

After reading about the evaluative setting, participants received more information concerning the commercial's creators (*authors* in the vignettes). Specifically, we described the creators of the commercial in terms of their *status* and their *social ties* to the experimental participants. We designed the manipulation of status by varying the creators' prestige and expertise. This manipulation was developed in line with the observation that expertise assessment is essentially "a status-organizing process" [54, p. 561] because individuals who have higher status are seen as more competent, whereas those who are of lower status are seen as less competent [55, p. 216]. In sum, in the status condition, the creators of the commercials were described as well-known experts in advertising, whereas in the no-status condition they were described as non-experts. The social ties manipulation was designed as *the presence or absence of a direct tie* to ensure consistency in the field study. Based on our manipulation, we informed the participants that they knew the commercial's creators and had collaborated with them in the past (i.e., direct ties condition), or that they did not know any of the commercial's creators and had never collaborated with them in the past (i.e., no-direct ties condition). Participants in the status and social ties condition read the description below (if assigned to the no-status and no-direct ties conditions, participants read the text in italics. Our scenario-based manipulations can be seen as a form of contextual priming, in which primed knowledge works as an anchor that is, as a standard of comparison for (re) evaluating the target, possibly resulting in contrasting the target away from the prime [56]. In following this approach, we relied on prior experimental studies using descriptive texts to prime status [e.g., 57, 58] or social ties [e.g., 59–61].

"In addition to the video, the organizers provide you with some information about the authors of the commercial. Looking at this information, you realized that all the authors of the commercial are **well-known experts** (*non-experts*) in advertising, and that you **know** some (*don't know any*) of them because you **collaborated with** them (*never collaborated with them*) on commercials in the past."

After reading the vignettes, participants in all four conditions watched and evaluated the same commercial on a new financial service. We selected this commercial from an actual Internet advertising contest where leading industry experts serve as judges in assigning various awards to commercials. The commercial chosen for the experiment was recognized as the *Best Computer: Software Online Video*. Link to the competition site: http://www.iacaward.org/iac/medium/Online-Video/best-online-video.html#. Link to the commercial site: https://www.youtube.com/watch?v=JHpVhEjufyA.

**Award propensity.** Our dependent variable measures whether participants are willing to assign an award to the commercial based on a 7-point scale (1 = "Definitely no", 7 = "Definitely yes"; the question was the following: "Would you assign an award to the commercial?").

**Manipulation checks.** We included both status and social tie's manipulation check. For the status manipulation check, we asked participants to answer the following question: "How much prestige do you think the authors have in advertising?" They rated the authors' prestige on a 7-point scale (1 = very low prestige, 7 = very high prestige). For the social ties manipulation, we asked the participant the following question: "How familiar do you feel with the authors?" Participants reported their answer on a 7-point scale (1 = not at all familiar, 7 = extremely familiar).

## Results and discussion

**Pre-analysis.** We first checked the presence of outliers for our dependent variable (award propensity) and identified nine outliers based on the Z-scores threshold of 2.5 SD [53, 62]. We removed these subjects from all subsequent analyses. We removed outliers at 2.5 SD in our experiments to increase the power by reducing the error variance. This is also the same reason why we removed inattentive participants by using the two attention checks and the commercial watching time. As Meyvis and van Osselaer [53, p. 1161] argue, the removal of participants "is often both legitimate and preferable. . .as this may produce both a more powerful and a more accurate test of the hypothesis."

**Manipulation checks.** First, we assessed whether the participants perceived the status manipulation by running a 2 (status: status vs. no-status) x 2 (social ties: direct ties vs. no-direct ties) between-subjects ANOVA on the rating of the creators' prestige. The analysis showed a significant main effect for status (F (1, 505) = 63.36, $p$<.001): participants in the *status* condition rated the commercial's creators as more 'prestigious' than participants in the *no-status* condition ($M_{status}$ = 4.68, $SD_{status}$ = 1.22; $M_{no\ status}$ = 3.77, $SD_{no\ status}$ = 1.36). No other significant effects were observed in the results. Similarly, to test the social tie manipulation, we ran a 2 (status: status vs. no-status) x 2 (social ties: direct ties vs. no-direct ties) between-subjects ANOVA on the rating of the creators' familiarity. The analysis showed a significant main effect for direct tie (F (1, 505) = 29.74, $p$<.001): participants in the direct ties condition perceived the commercial's creators as more 'familiar' than participants in the no-direct ties condition ($M_{direct\ ties}$ = 2.84, $SD_{direct\ ties}$ = 1.57; $M_{no\text{-}direct\ ties}$ = 2.12, $SD_{no\text{-}direct\ ties}$ = 1.38). No other significant effects were observed in the results. Thus, we concluded that the manipulations of our two independent variables worked as expected.

**Award propensity.** We ran a 2 (status: status vs. no-status) x 2 (social ties: direct ties vs. no-direct ties) between-subjects ANOVA on award propensity. Consistent with our field study, the results showed a significant two-way interaction (F (1,505) = 11.56, $p$ = .001). The main effects of status and social ties were not significant. In support of this finding, simple effects tests revealed that participants with direct ties to the commercial's creators were less willing to assign the commercial an award when creators with status (M = 4.37, SD = 1.15) rather than no-status (M = 4.81, SD = 1.13; F (1,505) = 7.86, $p$<.01) were involved. In contrast, participants with no direct ties to the commercial's creators were more willing to assign the commercial an award if status (M = 4.61, SD = 1.14) rather than no-status (M = 4.31, SD = 1.44; F (1,505) = 3.99, $p$<.05) creators were involved. Fig 4 graphs the lines, Fig 5 reports the bar charts and Table 5 reports the results.

These experimental findings corroborate the concomitant influence of status and social ties in shaping individual evaluative outcomes, thereby substantiating our results from the field study. The joint effect of status and ties is negative, confirming that audience members are less

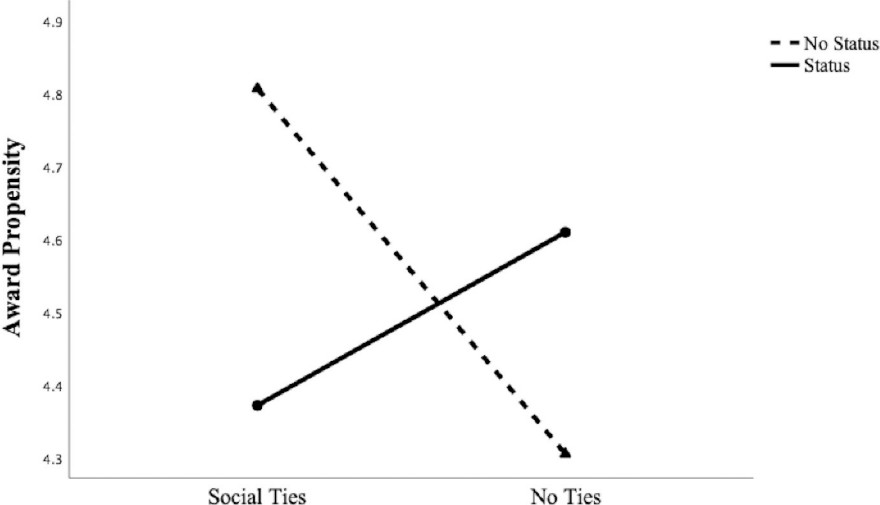

**Fig 4. Study 2: The effect of *status* and *social ties* on *award propensity*.**

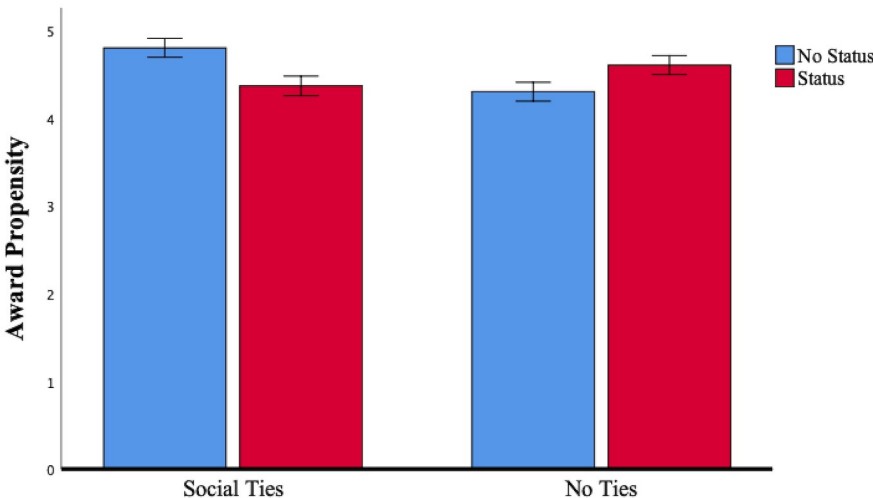

**Fig 5. Study 2: The effect of *status* and *social ties* on *award propensity*.** Note: Error bars are ± 1 SE.

**Table 5. Results for study 2.**

|  | Status | | No Status | | | |
|---|---|---|---|---|---|---|
|  | Social Ties (A) | No Ties (B) | Social Ties (C) | No Ties (D) | Interaction F (1, 505) | Simple Effects |
| Award Propensity |  |  |  |  |  |  |
| M | 4.37 | 4.61 | 4.81 | 4.31 | 11.56 ($p$ = .001) | B > D ($p$ = .046) |
|  |  |  |  |  |  | A < C ($p$ = .005) |
| N | 121 | 128 | 129 | 131 |  |  |

inclined to reward high-status candidates who are socially close to them. In the case of no ties, no recognition deterrent is present and the positive effect of status on award propensity prevails. Study 2 offers strong validation of the negative interaction effect because we manipulated status and social ties holding the project's quality constant. Doing so significantly mitigates the possibility that project level features may account for the effects observed in Study 1. In the *Appendix*, we report the results of a replication (experimental) study (i.e., Study 4) where we use a different manipulation of status ('famous' and 'not very famous' instead of 'well-known expert' and 'non-expert') and a different scenario describing the evaluative setting and we replicate the negative interaction between status and social ties. Please, see the *Appendix* for the full description and the results of this additional study.

## Study 3

While Study 2 increases the internal and external validity of Study 1's findings, it does not allow us to isolate the precise mechanism responsible for the negative interaction. Two equally plausible mechanisms could explain such an empirical pattern. The first is based on the understanding of status and social ties as substitutive judgment devices, namely the idea that social ties may substitute for status in conveying inferential information on evaluative targets, which in turn may guide evaluative decisions. To the extent that social ties channel private information on the evaluation target, audience members with ties to candidates who are in the consideration set should be less sensitive to status information. Conversely, the signaling effect of status should be significantly stronger for audience members who lack such ties and thus have no first-hand information on which to rely in their evaluation. The second explanation relies on social pressure arguments, namely on audience members' concern to be perceived as fair and disinterested in their evaluation since favoring candidates that are both high status and socially proximate to them can easily evoke suspicions of departure from the meritocratic ideal. The reflected glory evaluators enjoy through their connection to the winner of the tournament may predispose peers towards a morally problematic interpretation of those evaluators' motives. Thus, while both mechanisms could account for the same result patterning, the underlying explanations are profoundly different. In the first case, audience members who are socially close to the candidates arguably are less likely to use status information to reduce their evaluative uncertainty because social ties represent a more direct mechanism to temper evaluative uncertainty. In the second, audience members who are socially close to the candidates whose work they are expected to evaluate are less sensitive to their status because any further elevation of status of the target could exacerbate the perception that they are pursuing self-serving interests (even when this concern entails overruling a genuine assessment of merit).

In Study 3, we seek to unravel this duality by manipulating the evaluative context. In particular, we reasoned that if the interaction effect reflects evaluators' reputational concerns then the outcome of the evaluation should depend significantly on whether individual choices are private or in the public domain–and therefore subject to others' scrutiny. Note that in the field study the decisions of each jury member are collectively socialized and hence known to the other jury members. While this is often the case in peer-based evaluative settings in cultural fields (e.g., Cannes Film Festival, NSF evaluations), there are also evaluative settings in which decision-makers remain oblivious to each other's deliberations (e.g., Grammies, Oscars). Study 3 reproduces the previous studies as closely as possible; however, evaluators are explicitly told whether or not their evaluations are in the public domain (i.e., known to other evaluators). We then varied only the description of candidates' status, keeping constant their ties to evaluators. Accordingly, we asked all the study participants to evaluate a commercial created by peers with whom they were directly connected, and we manipulated both the status of the

authors of the commercial and the transparency of the evaluation process–i.e., whether or not evaluators' decisions are openly and collectively debated. Specifically, we developed two distinct descriptions of the contest: one in which the participants are told that their vote will be disclosed and the other in which participants are told that their vote will not be disclosed to the other jury members–which we label *public* evaluation and *non-public* evaluation, respectively. These manipulations allow us to ensure that only the participants asked to evaluate high-status peers in the public condition might be susceptible to stigmatizing perceptions. If the pressure to pre-empt potential reputational concerns shapes evaluative considerations, then the propensity to reward any given commercial should decline when the (socially proximate) author of the commercial is high-status and the evaluator's assessment is public. Stated differently, we should expect the probability to bestow an award on status peers–as opposed to no-status peers–to decline only when the vote is public. By contrast, when the vote is not in the public domain, we should not expect status peers to differ from no-status peers in terms of award propensity. If, on the contrary, status and social ties operate as substitute informational devices, then we should expect no difference between the public and the private conditions.

## Method

**Participants.**   A sample of five hundred and twenty participants was recruited via Prolific, an online UK-based platform [63]. The participants were compensated .70 GBP for completing the study. The recruitment was limited only to residents in the United Kingdom. As in the prior study, we used two attention check questions to exclude participants who did not pay attention while taking the survey. We used the same attention check of Study 2 to ensure that participants watched the commercial, and a similar IMC that instructed participants to leave the text-entry space blank for the following question: "In your opinion, which are the characteristics of good commercials? Please describe these characteristics below." Since in our vignettes we used the same commercial as in Study 1, we ensured consistency with the first experiment by including in the analysis only the participants who watched the commercial for more than 50 seconds and less than 122 seconds. As explained earlier, these methods are recommended to remove inattentive responses from online surveys to make sure that the pool of subjects is of high quality. The final sample consisted of 402 participants (68.8% female, $M_{age}$ = 35.58 years, 84.8% Caucasian).

**Material and procedure.**   We randomly assigned participants to one of the four conditions in a 2 (status: status vs. no-status) x 2 (public domain: public evaluation vs. non-public evaluation) between-subjects experiment. Participants were asked to read the same vignette used in Study 2, except for the information concerning the evaluation's public domain. In particular, to manipulate the public domain of the evaluation, participants in the public evaluation condition read: "Your **vote** will be **publicly** disclosed to the other jury members to collectively select the winner." On the other hand, in the non-public evaluation condition, participants read: "Your **vote** will **not be** disclosed to the other jury members to collectively select the winner." Like in Study 2, participants received specific information regarding the commercial's creators. In all four conditions, we held direct ties between the creators of the commercial and the experimental participants constant by telling the participants that they knew the commercial's creators and had collaborated with them in the past. We then manipulated the status of the creators by applying the same manipulation as in Study 2. Finally, participants were asked to watch and evaluate the commercial already employed in Study 2.

**Award propensity.**   The same question from Study 2 was used to measure the propensity to award the commercial.

**Manipulation checks.** We included both status and public domain manipulation checks. Consistently with Study 2, we used the same manipulation check for status by asking participants how much prestige they think the authors have in advertising on a 7-point scale (1 = very low prestige, 7 = very high prestige). For the public domain's manipulation, we asked the participant the following question: "Do you think the award decision is anonymous?" Participants reported an answer on a 7-point scale (1 = definitely no, 7 = definitely yes).

## Results and discussion

**Pre-analysis.** Following the same approach of Study 2, from an analysis of outliers on our dependent variable–award propensity–we identified five outliers based on the Z-scores threshold of 2.5 SD [62, 53]. We removed these subjects from subsequent analyses.

**Manipulation checks.** We tested the effectiveness of the status manipulation with a 2 (status: status vs. no-status) x 2 (public domain: public evaluation vs. non- public evaluation) between-subjects ANOVA on the rating of the creators' prestige. The result revealed a significant main effect for status (F (1, 393) = 174.23, $p<.001$): participants in the status condition considered the creators of the commercial more prestigious than participants in the no-status condition ($M_{status}$ = 5.34, $SD_{status}$ = 1.04; $M_{no\ status}$ = 3.69, $SD_{no\ status}$ = 1.46). The analysis also showed a significant main effect for public domain (F (1, 393) = 4.29, $p$ = .039; $M_{non-public}$ = 4.61, $SD_{non-public}$ = 1.44; $M_{public}$ = 4.46, $SD_{public}$ = 1.57) and a marginally significant interaction (F (1, 393) = 3.84, $p$ = .051). Since these results suggest a potential confound in our experimental manipulation, we followed Perdue and Summers' [64, p. 323] recommendation and compared the effect sizes of each factor to check the validity of our manipulation. The effect size of the status manipulation ($\eta^2_{status}$ = .307) was 28 times greater than the effect size of the public domain's manipulation ($\eta^2_{public\ domain}$ = .011) and 31 times greater than the effect size of the interaction effect ($\eta^2_{intercation}$ = .01), suggesting that our status manipulation worked as expected. To test the public domain's manipulation, we ran a 2 x 2 ANOVA on a decision's anonymity. The analysis showed a significant main effect for public domain (F (1, 393) = 389.71, $p<.001$): participants in the non-public condition perceived the decision to be anonymous compared to the public condition ($M_{non-public}$ = 5.28, $SD_{non-public}$ = 1.54; $M_{public}$ = 2.28, $SD_{public}$ = 1.48). We found no other significant effects on the results. Thus, our public domain's manipulation was effective.

**Award propensity.** A two-way ANOVA on award propensity revealed no significant main effect for status (F (1, 393) = 2.24, $p>.05$), public domain (F (1, 393) = .024, $p>.05$.), and the interaction (F (1, 393) = 1.75, $p>.05$). Yet, more importantly, the results of planned contrasts analysis showed the expected pattern. Meyvis and van Osselaer [53] argue that "requiring authors to demonstrate a reliable main effect before allowing the testing of planned contrasts (which more precisely test the hypothesis) is not a statistically sound argument" [53, pp. 1171–1172]. Also, Keppel and Wickens [65] state that "when an experiment has been designed to investigate a particular planned contrast, it should be tested regardless of the significance of the omnibus F statistics" [65, p. 116]. Since we did not expect some of the experimental conditions to differ from each other, the planned contrasts analysis is more precise and powerful than the omnibus F test.

First, we examined the effect by comparing the "status and public condition" vs. the "no status and public condition." As expected, the planned contrast revealed that participants were less likely to assign an award to a commercial created by status peers ($M_{status\ and\ public}$ = 4.25, $SD_{status\ and\ public}$ = 1.17) compared to a commercial created by no status peers when their vote was public ($M_{no\ status\ and\ public}$ = 4.58, $SD_{no\ status\ and\ public}$ = 1.12; $t$ (393) = -2.00, $p<.05$). To test the other planned contrast, we compared the "status and non-public condition" to the "no

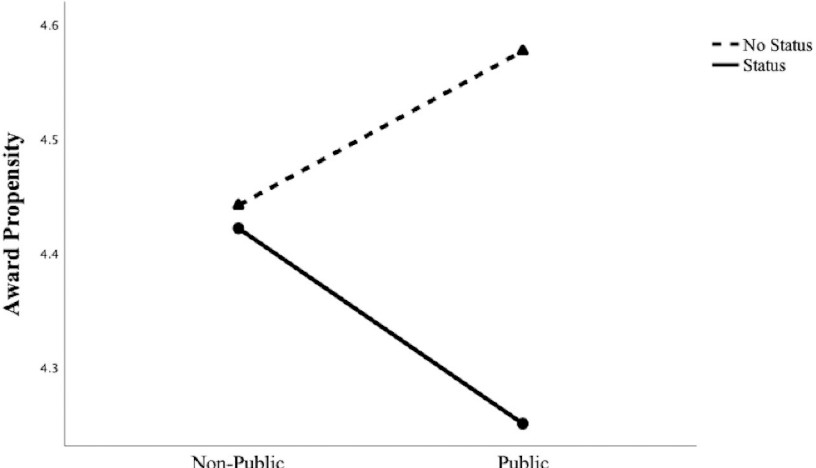

**Fig 6. Study 3: The effect of *status* and *public domain* on *award propensity*.**

status and non-public condition." As expected, this contrast was not significant, $t$ (393) = -.12, $p > .05$ ($M_{\text{status and non-public}} = 4.42$, $SD_{\text{status and non-public}} = 1.17$, $M_{\text{no status and non-public}} = 4.44$, $SD_{\text{no status and non-public}} = 1.14$). Overall, the contrast tests confirmed previous findings, further supporting our speculation that a stigma deflection effect is at play because a commercial's likelihood to receive an award declined significantly only when jurors evaluated the work of status peers directly connected to them and their vote was public. We observed no such effect when the evaluation occurred privately. Although we cannot conclusively rule out the alternative explanation that status and social (ties) information operate as substitutive information devices, we believe that this alternative explanation is unlikely to affect the results. Fig 6 graphs the lines, Fig 7 reports the bar charts and Table 6 reports the results.

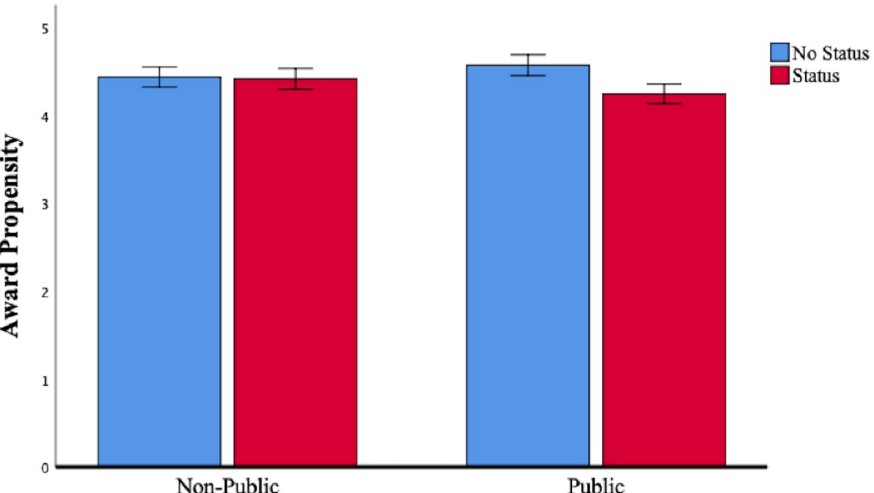

**Fig 7. Study 3: The effect of *status* and *public domain* on *award propensity*.** Note: Error bars are ± 1 SE.

**Table 6. Results for study 3.**

| | Status | | No Status | | |
| --- | --- | --- | --- | --- | --- |
| | **Non Public (A)** | **Public (B)** | **Non Public (C)** | **Public (D)** | **Planned Contrasts** |
| Award Propensity | | | | | |
| M | 4.42 | 4.25 | 4.44 | 4.58 | B < D ($p$ = .047) |
| | | | | | A < C (n.s.) |
| N | 95 | 108 | 102 | 92 | |

## Discussion and conclusions

In the sociological and organizational literature, tournament rituals operate by selectively allocating recognition among competing candidates [66, 67]. Well-known examples of ceremonies in the cultural domain that signal creative achievement epitomizing peer-based recognition are the Academy Awards in motion picture [68], the Grammies in music [67], the John Bates Clark Medal in economics [69], the Nobel Prize for advances in culture and science [70], and so on. Operating as markers of distinction, these ceremonies shape a cultural field's status ordering [71]. As such, they have received significant attention from scholars interested in understanding the socio-cognitive mechanisms underlying these evaluative efforts [72–74, 21, 75, 25]. Prior research, therefore, has shown how status and social networks are pervasive forces that produce and reproduce attributions of distinction in artistic and scientific evaluative settings. Although several studies have documented the importance of status or social networks across a variety of domains of cultural production, very limited research has focused on how the two mechanisms operate in tandem to shape evaluative outcomes.

We argued that proximity in the social network could moderate the almost universal association between status and recognition. Specifically, social ties to members of the evaluating audience could increase or decrease the positive effect of the candidate's status. In a field study of award conferrals in advertising contests, we found social proximity between advertising professionals and jury members to attenuate the positive association between status and recognition. Conversely, social ties between audience members and candidates were particularly beneficial in fostering the recognition of candidates who lacked status credentials. Supplementary online experiments aimed at further corroborating this finding and clarifying the underlying mechanism confirmed the negative interaction effect. It is indeed reassuring that we observe the same pattern of effects in the real world (Study 1) and in the online experiments (Studies 2 and 3), which supports the internal and external validity of our results–an important criterion for impactful research in the social sciences [76].

However, Studies 1 and 2 cannot distinguish between two slightly different mechanisms that could underlie the negative interaction effect. One possibility–rooted in the classic understanding if social ties as channels of private information–is that social proximity reduces the saliency of status as an information device. The second possibility is that participants rely less on status when they try to project an image of disinterestedness by not rewarding high-status candidates with whom they have a history of collaboration. To understand whether there is more in the interplay of status and ties than a compensative flow of information, in Study 3 we manipulated the nature of the evaluative context. As in Study 2, we found that status reduces the probability of bestowing an award on socially proximate candidates, but only when participants are induced to think that their decisions are public–i.e., they have to justify their evaluation before other jury members. Conversely, when the evaluation is private, the negative effect of status disappears. Since this result is obtained by keeping the audience-candidate social tie constant across different conditions, it is plausible to conclude that reputational concerns–as

opposed to inferential information–shape evaluators' decisions [77]. While we can only speculate on the exact nature of this concern, the observation of differential effects across the public/non-public conditions is strongly evocative of Goffman's [78] "front stage"–"back stage" tension that may envelop audience members' evaluative choices. When decisions are in the "front stage," the perception that evaluators are pursuing implicit personal gains rather than adhering to disinterested rules and practices is likely to elicit efforts to project such disinterestedness and hence deflect attention away from any signal that would render their choices overly susceptible to sceptical scrutiny. In our case, we speculate that insofar as social proximity heightens exposure to public criticism for the alleged pursuit of self-serving interests, audience-candidate relationships that may be publicly perceived as structuring the awarding process may also dampen the signalling saliency of status. The reason is not that the private information channelled via a social relationship replaces the information conveyed by the candidate's status, but that such a relationship "influences how others perceive the actor" [39, p. 563], thereby lowering the evaluator's permeability to status cues. This presumption seems particularly pertinent to ostensibly meritocratic cultural settings characterized by a strong vocational drive and professional ethos [79], where the potential stigma stemming from the alleged transgression of the meritocratic ideal may be particularly severe for one's reputation. Future experimental research more geared toward the analysis of the evaluators' decision-making patterns and other micro intervening processes underlying our effects is needed to probe the plausibility of this interpretation.

The findings of this study are important because they shift the emphasis away from either status or social network explanations for recognition and refocus the attention on how these two mechanisms combine in the creation of prestige hierarchies and how such a combinatory effect may itself vary with the specific relational context in which the evaluative activity occurs. Specifically, the negative interaction effect between status and social ties has interesting implications for the dynamics of cumulative advantage. The finding that marginal returns to status diminish with social network proximity adds another piece of evidence that there might be endogenous constraints on the Mathew effect [80]. It is often assumed that the self-reinforcement of status and networks inevitably leads to "winner-take-all" dynamics as cumulative benefits accrue mostly to those who, even by small margins, are in superior positions [81, 82]. This assumption needs to be qualified especially if, under certain conditions, status considerations may lose saliency in the eyes of audience members in charge of relinquishing material and symbolic resources to competing candidates. Further research should elucidate which conditions are worthy of future inquiry.

We believe that our paper takes an important step towards a more precise articulation, in both theoretical and empirical terms, of the role of evaluating audiences in explaining status-based recognition mechanisms. A better understanding of how audiences shape status dynamics is important to mitigate the tension between achievement and ascription that is at the core of meritocratic evaluative settings–whereby audiences are supposed to justify their deliberations based on standards that can be articulated independently of available options [11]. In addition, understanding how audience evaluations may change with the degree of scrutiny to which they are amenable seems crucial in light of ever-increasing calls for transparency and fairness in public life [19]. It is therefore worthwhile to probe the interaction between candidates' status and audience-candidate connectivity, particularly if one considers that social proximity between producers and audience members is a constitutive feature of peer-based evaluative settings. Because of the role-switch structure of these settings, audience members are also members of the same community as the candidates they evaluate, even though they take on different roles [83], and so–more often than not–they may have few degrees of separation from each other [22, 84].

More generally, our findings speak to prior work attentive to the role of the social-relational context in shaping assessments of merit. Ridgeway and Correll [85] consider a social-relational context any situation in which an actor has to take into account the expected reactions of others in determining how to act, because such reactions will have consequences for her interests. Other lines of scholarship point to how personal preferences often seem to fade in salience relative to what is publicly endorsed in a status hierarchy [11, 12]. This phenomenon is especially apparent in research on "politics of dissimulation"–e.g., Norbert Elias' [86] scholarship on authoritarian systems–whereby public displays of allegiance to the official credo may mask a great deal of private disagreement. Findings in experimental economics also suggest that individuals tend to change their strategic choices depending on whether they are isolated or part of a group, and on whether there is an audience observing their choices [20]. Likewise, research on social evaluation [87], as well as studies on the social transmission of ideas of fairness [88, 89], show that the apparent social validity of something (e.g., social norms on pro-social behaviour) shapes individual decision-makers' allocative choices, independent of their personal assessments of quality or fairness. Our study adds conceptual and empirical nuance to these lines of research by revealing how the influence of crucial social cues–such as status and ties–on evaluators' choices may depend on whether members of the evaluating audience can infer key motivational premises from those choices. In light of recent findings revealing the conditions under which social norms in collective decision-making settings may play a smaller role than previously assumed [90, 91], the extent to which socio-normative pressures–as opposed to an individual's professional ethos, concern for her image or other behavioural mechanisms–are indeed driving this interaction effect could be an exciting area for future inquiry. To this end, we might posit a variety of peer-based evaluative contexts from those akin to our setting, where individual choices are collectively socialized and visible to other decision-makers (e.g., Cannes Film Festival, National Science Foundation), to those where individual choices remain private (e.g., Grammies, Academy Awards).

Several questions merit further attention. Our data do not capture the process by which jury members collectively make their decisions, in particular how their (often) conflicting opinions are reconciled and consensus on which projects to reward is reached. The combination of archival and interview data offered some suggestive insights, but the use of an ethnographic approach would be better suited to gain a more nuanced understanding of the processes by which reward allocation decisions are collectively made. One could then probe more deeply the conditions under which the desire to distance oneself from morally dubious evaluations is, in fact, driving those decisions, independently from the evaluator's actual personal beliefs. Another issue is that prior project collaborations capture only a subset of relevant audience-candidate interpersonal relationships. While we endeavoured to control for other possible manifestations of these social intercourses, an extension could be to supplement project collaboration data with additional data on more informal interactions (e.g., advice, friendship, mentorship, and so on) and examine whether patterns of rewards allocation can then be explained more accurately. Along this line, another interesting research direction would be to compare and contrast the role of formal vis-à-vis informal ties in shaping evaluative outcomes for different levels of status. These are but some of the many questions that future research could explore in greater depth.

## Supporting information

**S1 Table. Results for study 4.**
(DOCX)

**S1 Fig. Study 4: The effect of *status* and *social ties* on *award propensity.***
(DOCX)

**S2 Fig. Study 4: The effect of *status* and *social ties* on *award propensity.***
(DOCX)

**S1 Appendix.**
(DOCX)

**S1 Data.**
(ZIP)

## Author Contributions

**Conceptualization:** Gino Cattani, Simone Ferriani.

**Funding acquisition:** Simone Ferriani.

**Methodology:** Erik Aadland, Denise Falchetti.

**Project administration:** Gino Cattani.

**Writing – original draft:** Erik Aadland, Gino Cattani, Denise Falchetti, Simone Ferriani.

**Writing – review & editing:** Erik Aadland, Gino Cattani, Denise Falchetti, Simone Ferriani.

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
