## [Decision Letter · Decision Letter 0]

18 May 2020

PONE-D-20-09850

The Interplay Between Status and Social Proximity in Peer Evaluation: A Socio-Cognitive Perspective on Reward Allocation

PLOS ONE

Dear Prof. cattani,

Thank you for submitting your manuscript to PLOS ONE. After careful consideration, we feel that it has merit but does not fully meet PLOS ONE’s publication criteria as it currently stands. Therefore, we invite you to submit a revised version of the manuscript that addresses the points raised during the review process.

We would appreciate receiving your revised manuscript by Jul 02 2020 11:59PM. To enhance the reproducibility of your results, we recommend that if applicable you deposit your laboratory protocols in protocols.io, where a protocol can be assigned its own identifier (DOI) such that it can be cited independently in the future. For instructions see: http://journals.plos.org/plosone/s/submission-guidelines#loc-laboratory-protocols

We look forward to receiving your revised manuscript.

Kind regards,

Valerio Capraro

Academic Editor

PLOS ONE

Journal Requirements:

2. We noticed you have some minor occurrence of overlapping text with the following previous publications, which needs to be addressed:

* Cattani et al, “Friends, Favours and Cliques: Relational Mechanisms of Recognition in Peer-Based Tournament Rituals” Academy of Management Journal, 2019, 62(3): 883-917.

* Correll, Shelley J., et al. "It’s the conventional thought that counts: How third-order inference produces status advantage." American Sociological Review 82.2 (2017): 297-327.

In your revision ensure you cite all your sources (including your own works), and quote or rephrase any duplicated text outside the methods section. Further consideration is dependent on these concerns being addressed

4. Thank you for stating the following financial disclosure: "15302"

"no"

8. Please amend the manuscript submission data (via Edit Submission) to include author Erik Aadland and Denise Falchetti.

Additional Editor Comments (if provided):

The reviewers are positive towards the paper, but they have several suggestions for a revision. Therefore, I would like to invite you to revise your work following the reviewers' comments. Also, I would like to add one comment related to your discussion of references 83 and 84, where you say that "studies on the social transmission of ideas of fairness [83; 84] show that the apparent social validity of something (e.g., social norms on pro-social behaviour) shapes individual decision-makers’ allocative choices, independent of their personal assessments of quality or fairness". I would like to encourage you to be a bit more careful in this regard, as we have recent results showing that dictator game behaviour is primarily driven by personal norms (Capraro and Vanzo, Judgment and Decision Making, 2019) and that in allocation tasks where the personal norm conflicts with the descriptive norm people are more likely to follow the personal norm (Capraro and Rand, Judgment and Decision Making, 2018); this suggests that social norms might play a smaller role than previously assumed.

I am looking forward for the revision.

Reviewers' comments:

Reviewer's Responses to Questions

**Comments to the Author**

1. Is the manuscript technically sound, and do the data support the conclusions?

Reviewer #1: Partly

Reviewer #2: Yes

Reviewer #3: Partly

2. Has the statistical analysis been performed appropriately and rigorously? 

Reviewer #1: Yes

Reviewer #2: Yes

Reviewer #3: Yes

3. Have the authors made all data underlying the findings in their manuscript fully available?

Reviewer #1: Yes

Reviewer #2: No

Reviewer #3: Yes

4. Is the manuscript presented in an intelligible fashion and written in standard English?

Reviewer #1: No

Reviewer #2: Yes

Reviewer #3: Yes

5. Review Comments to the Author

Reviewer #1: Summary

This paper studies peer-based evaluative settings. It focuses on situations in which peers have to evaluate each other for the attribution of an award. The authors investigate how social proximity (between the evaluator and the candidate for an award) and the status of the candidate interact and how they jointly affect the chance that the candidate receives the award. To that end, the paper relies on 3 studies. The first is a correlational study which relies on a data collected in the context of a digital advertising contest. They show that social proximity and status negatively interact: For a high-status candidate, the chance of getting the award decreases if his/her social proximity with the evaluation committee increases. For low-status individuals, social proximity increases the chance of getting the award. They replicate this result in a vignette study run on MTurker with a 2x2 design in which status and social proximity are manipulated (Experiment 1). In a second vignette study ran on Prolific (Experiment 2), they vary whether the decision of a jury member (the decision maker) is public or private. They show that the social-proximity tempers the benefits of having a high-status only when the decision is public, but not when it is private.

Assessment

I enjoyed learning about these findings. Overall, I think this is an interesting research question, that the authors addressed rather carefully. I very much appreciate the combination of observational data with follow-up experiments. I do have some concerns about the current version of the paper, which mostly relate to writing (the paper is too long!), the analysis of the field data, and the interpretation of the findings. However, I think all these points can be addressed by the authors. One last concern I have is that the study uses deception, which is something that our profession tries to prevent by all means.

Main Comments

Writing. I think the paper is too long compared to the message it delivers. There are too many unnecessary details in the main text in which the read gets lost, which ultimately leads him to loses the big picture. For example, the discussion on the construction of the social ties/project status variables (p.12), the discussions on “Pre-analysis” and the long “Manipulation checks” sections for Study 1 and 2, the “Methods” discussions in which you go as far as telling which stata command you used, the discussion of “interviews with key informants”. I think these kind of details should be either in footnotes or in Appendix. This would make the text MUCH smoother to read. I also didn’t find the section “Overview of studies” particularly useful.

The field study. This largely relates to my comment on writing, but I think the paper could benefit a lot from integrating the field study into the introduction. In a sense, the field study is just here to document an empirical pattern that the authors then explain using Experiments 1 and 2. For what it’s worth, I think that the section of the field study is way too long (again, so much unnecessary details). What I would suggest is that the authors find a way to limit their discussion of this field study to what is needed to understand the context, and just explain Figure 1 and Figure 3 in the text (these are cool figures that tell everything to the reader!). This is all we need to understand that there is a strong interaction.

Analysis of field study. Why don’t you simply use a simple multinomial ordered logit? Isn’t this something that is more common than the method you currently use? I also find that your discussion of the regression table is a bit superficial (here: more details would be good!). Everyone can see that the value of a coefficient is X, but what would be useful is that you deliver an economic interpretation for these coefficients. What do they mean? And what are they? Are they odds-ratios? Are they marginal effects? Do they have a meaningful interpretation? Because in many cases (e.g. with probit/logit models), coefficients cannot be interpreted; only marginal effects can. In any case, I would suggest you mostly focus on Figure 1 and 3 to discuss the results of the field study: they contain all the information we need (Note: It is confusing in Table 4 to have all the control variables at the top of the table, and the coefficients we care about at the bottom.)

Information acquisition: You often speak about how peers can use status/social-proximity to get information on the candidates. For example, on p. 5, you say “the existence of social ties between audience members and candidates should reduce the saliency of status as a signal of the quality of candidates and their work.” You also talk about this on top of p.23. I agree that this might be an important dimension. However, I don’t think that your study can say anything about that (because, in your two experiments, jury members don’t have any private information about candidates). In a sense, your decision cuts off this channel, since by construction private information cannot play a role in your experiment. I therefore would suggest you do not speak about this with such details, but instead focus on the alternative interpretation which is that people might not like to appear to favor someone with high status to which they are socially close if the decision is public. I would, however, mention that a nice feature of your design is that it turns the “private info” channel off.

Deception. Both experiments use deception. In Study 2, you tell subjects that “the jury selects the winner collectively by disclosing the vote cast by each jury member” although this will never happen. In Study 3, it is even worse as you explicitly tell your subjects that “your vote will be publicly disclosed to the other jury members” in the public condition. I don’t think that this is something that was necessary (you could have run the experiment, either online or in the lab, without deception). Economics has a strong norm against deception, and I think this discussion applies to online studies as well. (Just like a lab loses its reputation when it deceives subjects, MTurkers know and discuss in online forums which HITs/requesters use deception.)

Interpretation of results. I think you over-interpret a little bit your results. For example, in the conclusion you say “we demonstrated that when audience members do not have to justify their decisions before other members […]” and “social proximity tempers the effect of status on candidates’ recognition when decisions are in the public domain and, therefore, potential violations of the meritocratic ideal in social evaluation are more likely detected and stigmatized, if not punished.” In my opinion, this is quite speculative. An alternative explanation might be that people care about their social image and don’t want to be perceived as someone who favors socially-close candidates. Your conditions cannot really tease out the mechanisms why the public condition leads to these results, so I would be a little bit more modest when interpreting them. (This comment also applies to the introduction.)

More minor comments

p.3: It would be useful to define status at the very beginning of the text.

p.3: “they are usually granted more recognition for their performance relative to low-status actors for equivalent performance” : do you have a reference for that?

p.5: I would use “could” instead of “should” when formulating the hypotheses.

p.5: I do not understand your explanation for why positive effect of status should get even stronger when high-status candidates have ties to members of the evaluating audience.

p.5: I don’t think the example on dyadic business relationships fits very well to your context.

p.6: Why don’t you describe a bit more in the intro the context of the field study and what you find?

p.15: Why use past tense to describe the coefficients obtained in the field study (e.g. “the coefficient for project status was 1.062”) ?

In Experiments 1+2, did your manipulation checks come before or after the measurement of your main outcome? I think in your case, there might be important demand-effects/desire for consistency.

I find your labeling of the different studies very confusing. Either say Field study + Online Experiment 1 and online Experiment 2, or say Study 1-2-3. But please be consistent.

In your two vignette studies, how are treatment assigned: between-subjects, or within? I could not find this information.

Avenue for future research (p.33). This is a matter of taste, but I would not keep this paragraph.

Reviewer #2: I commend the authors for all the work they have done to incorporate the suggestions of the esteemed fellow reviewers. I only have a couple of minor requests for clarification, mostly to improve the clarity of exposition.

- P.5 "…render neighborhoods synonymous with the activity of those who win them" could be edited for clarity.

- P. 6:" de facto the recipients' status boost into their own stigma," could be edited for clarity.

- Table 2: the mean of "Conflict of interest" is puzzling. Were about half of the evaluators in a conflict of interest? Were these people then sitting in juries they should not have sit in according to contest rules?

- P. 18: "well-controlled lab study," you use MTurk so... no lab. Also, the well-controlled can probably go. The "lab" reappears at p. 30.

- I do not want to make a big fuss about the limits of vignette studies: the other reviewers already pointed those out. However, statements like "participants in the direct ties condition rated the commercial's creators as more familiar" (p. 22) are a little awkward. Similarly, when you say that "the manipulations… were successful" (id.) I think what you mean to say is that your participants buy into what you told them in the instructions. It is an attention check.

- P. 23: "These experimental findings offer causal evidence." I doubt you have established causation through analysis of variance. I think what you mean to say is that a different methodology corroborates the findings of your field study 1.

- P. 24: "disinterestedness ideal." I would stick to the "meritocratic ideal, the term you used earlier in the paper.

- P. 24: "The second explanation relies on reputational arguments…" According to this explanation, giving a prize to a buddy who is also a celebrity in the field looks bad. I would say that giving a prize to a friend will probably look bad (especially if there are quality concerns). But status and recognition in the field do not undercut reputation by a comparable magnitude to proximity. Think about the refereeing process. There is much attention to conflicts of interests that involve personal ties. I view this as a failsafe for those whose internal regulator does not work to decline refereeing requests for papers written by close friends. But no one thinks that mere prominence in a field is at odds with the "disinterestedness ideal." Otherwise, we should prohibit every author from an R1 institution from submitting a paper (God forbid Nobel Prize winners!). That sounds a little extreme a conclusion. I would be curious to hear what the authors think. The authors make the same point at the end of page 30 and at page 6.

- Same page: "In the second, audience members who are socially close to the candidates whose work they are expected to evaluate are less sensitive to their status…" I would say that if you are socially close to someone, then you should make an extra effort at evaluating the project impartially. Or perhaps, if you want to project an image of extreme resilience to social proximity considerations, you should award the prize with lower probability. I do not understand why reputation concerns command that you should disregard someone's status when the contestant is proximate.

- P. 25: "These manipulations allow us to ensure that only the participants asked to evaluate (high?) status peers in the condition are (I would say might be-- again, it is a vignette) susceptible to reputation concerns."

- P 25: just before the Methods, I would add that the other explanation, that status loses information value when you are proximate, does not lead you to predict that publicity, or lack thereof, would make a difference.

- P. 25. High data quality? Do you mean that the pool of subjects is of high quality?

- P. 29: "Paramount…the roles of status and social networks…" I believe it is better to say status and social proximity. Status is proxied as network centrality in your paper. Same on page 3.

- P. 31: The quote attributed to Dimant that "behavior that 'benefits oneself at the expense of others is socially harmful…" can go. You do not engage in welfare analysis in the paper. And the quotation reflects a pre-Coasean point of view on causation. If I consume an apple, I am benefiting myself at the expense of someone who is not eating the same apple. Dimant’s first best world, in which there are no reciprocities in resource use, is a place no one has ever inhabited.

Warmest regards

Reviewer #3: Report for „ The Interplay Between Status and Social Proximity in Peer Evaluation: A Socio-

Cognitive Perspective on Reward Allocation “

Manuscript ID: PONE-D-20-09850

SUMMARY

The authors test the role of social status and social proximity for evaluation and recognition choices such as the allocations of awards with field data and two experiments. Based on the literature, they ask if status of the candidate and social proximity between candidate and jury members act as substitutes or complements in how they combine to produce evaluative outcomes. Based on observational data from an award in advertising industry in Norway and two experiments, they show a negative interaction between status and social proximity. The experiment suggests that when evaluations are public, social proximity tampers the effect of status on recognition, but not when decisions are private. Thus, reputational concerns of the jury members may lead to a substitutional effect in their setting.

RECOMMENDATION TO THE EDITOR:

I recommend a revise and resubmit. I think the topic is very interesting and relevant and the combination of field data and an experiment is interesting and promising. Unfortunately, the design of the experiments has many problems, which make me unsure whether the paper should be accepted for publication without major changes in the design and possibly even a replication of the experimental studies. Especially, I am not convinced by the treatment manipulations and by the interpretation of the behavior they observe, see below. However, I still would like to give the authors a chance to resubmit a modified version and explain their choices.

MAIN COMMENTS

I don’t know this precise field of literature well, so I would leave it to another referee to judge the novelty and originality of the contribution this study makes to the existing literature. If the empirical/experimental evidence is sufficiently novel, I think the topic is very relevant and interesting in many areas and professions. One application certainly is peer review in academia.

STUDY DESIGN:

• It’s a nice feature of the article, that the authors try to establish additional external validity and try to validate their own interpretation of their findings by interviewing professionals in the field.

• FIELD STUDY: Can the authors elaborate a bit more on how project quality is controlled for? It seems like this is not perfectly captured by their variables and it is very likely to be correlated to project status. This might be a problem for the identification in the field setting.

• EXPERIMENT 1:

o I find it a bit problematic that the vignettes only contain information about creators social status and social ties (p.20). This makes the situation highly hypothetical and also can create experimenter demand effects.

o The social ties condition is highly abstract--“imagine you have collaborated with them in the past”—and does not really induce true social ties. There are many experiments that truly manipulate “social proximity”, i.e. that really induce a social distance or connection but priming them on common preferences, creating groups. It would have been even possible here to actually have subjects work together on a task and then match them for the jury-candidate experiment… There are many good examples in previous studies and I don’t understand why the authors don’t use them. I’m not convinced this manipulation has the desired effect.

o DV (p.21): Subjects watch one commercial and should then say how likely they are to award it. How should they make this choice? They have nothing to base their choice on, i.e. no other commercials to compare it to. Because of this, the manipulations by the experimenter might actually have a larger effect than they would have otherwise.

• EXPERIMENT 2:

o The dampened signaling effect of status if a jury member and a candidate have close social ties does NOT work in their design because they HAVE NO actual social ties. This mechanism cannot be detected in their experiment (p.24). This also shows that their manipulation of social proximity is highly problematic for their purpose (see above).

ADDITIONAL MINOR COMMENTS

• In Experiment 1, the authors mix “status” and “expertise” in their treatment manipulation which I find very questionable. What is the reason for doing this? It seems easy to disentangle the two and therefore to avoid this confound. Giving an “expert cue” and interpreting its effect as “status” seems like an interpretation that is probably not warranted.

• Especially without pre-registration, removing any subjects from the analysis is questionable, e.g. removing the outliers, p.22.

• Overall, I recommend to shorten and streamline the text.

• Tables in the text instead of in the appendix would have helped a lot to make the paper more readable.

6. PLOS authors have the option to publish the peer review history of their article (what does this mean?). If published, this will include your full peer review and any attached files.

Reviewer #1: No

Reviewer #2: No

Reviewer #3: No

---

## [Author Response · Author response to Decision Letter 0]

21 Jul 2020

Reply to Editor

We appreciate the time and thought you have put into your evaluation of our manuscript. Your comments highlighted areas for improvement and, below, we explain how we responded to your comments. Specifically:

1) We streamlined the paper and moved some details into either a footnote or the Appendix.

2) We clarified several points that the reviewers found confusing.

3) We re-estimated the econometric models (Study 1) by conducting several new robustness checks (as described on p. 17). In particular, we tried to address the reviewers’ questions regarding project quality and the choice of model specification.

4) We toned down some of the claims, especially when we interpret and draw implications from the results in the discussion section.

5) We ran a new experiment (described in the Appendix) as a robustness check.

6) We included some new references that helped us flesh out our theoretical arguments, interpret the results and define more precise scope conditions for them.

7) We changed the title of the manuscript.

We hope you will find this revised version of our manuscript to deserve publication in PLOS ONE. Again, thanks for your guidance throughout the review process. We remain at your disposal for any further clarification.

 

Reply to Reviewer #1

We very much appreciate the time and thought you have clearly put into evaluating our manuscript. Your comments have highlighted several areas where we could improve our paper. Below, we explain how we have responded to them. For the sake of convenience, we present your comments first and then our response (in bold). Please, note that we changed the title of the paper.

I enjoyed learning about these findings. Overall, I think this is an interesting research question, that the authors addressed rather carefully. I very much appreciate the combination of observational data with follow-up experiments. I do have some concerns about the current version of the paper, which mostly relate to writing (the paper is too long!), the analysis of the field data, and the interpretation of the findings. However, I think all these points can be addressed by the authors. One last concern I have is that the study uses deception, which is something that our profession tries to prevent by all means.

Thanks!

Writing. I think the paper is too long compared to the message it delivers. There are too many unnecessary details in the main text in which the read gets lost, which ultimately leads him to loses the big picture. For example, the discussion on the construction of the social ties/project status variables (p.12), the discussions on “Pre-analysis” and the long “Manipulation checks” sections for Study 1 and 2, the “Methods” discussions in which you go as far as telling which Stata command you used, the discussion of “interviews with key informants”. I think these kind of details should be either in footnotes or in Appendix. This would make the text MUCH smoother to read. I also didn’t find the section “Overview of studies” particularly useful.

In revising the paper, we tried to shorten the paper and move some details to either new footnotes (footnotes 2, 3, 4, 5, 11 and 12) or the Appendix. We also streamlined a bit the overview of studies section. At the same time, we would like to emphasize that we had to maintain some of the details (e.g., with respect to the variables in the field study) to address the concerns of a previous reviewer who wanted us to explain the variables more extensively and in greater depth.

The field study. This largely relates to my comment on writing, but I think the paper could benefit a lot from integrating the field study into the introduction. In a sense, the field study is just here to document an empirical pattern that the authors then explain using Experiments 1 and 2. For what it’s worth, I think that the section of the field study is way too long (again, so much unnecessary details). What I would suggest is that the authors find a way to limit their discussion of this field study to what is needed to understand the context, and just explain Figure 1 and Figure 3 in the text (these are cool figures that tell everything to the reader!). This is all we need to understand that there is a strong interaction.

Thank you for your kind guidance and encouragement to improve the fluency of the study. We streamlined Study 1 (the field study) as much as possible. We removed some descriptive material, moved some sections in the footnotes and relocated contextual information in the Appendix. However, we would also like to stress that the field study is crucial to demonstrate that there is indeed a negative interaction effect operating in the context of digital advertising awards allocation in Norway. As such, it is instrumental in establishing the external validity of the interaction effect. As currently presented in the paper, the field study conveys the empirical patterns in a way that enables other scholars to assess the merits of the field study and test the patterns. We adhere to this fundamental scientific principle in the reporting of our investigation’s results.

Analysis of field study. Why don’t you simply use a simple multinomial ordered logit? Isn’t this something that is more common than the method you currently use? I also find that your discussion of the regression table is a bit superficial (here: more details would be good!). Everyone can see that the value of a coefficient is X, but what would be useful is that you deliver an economic interpretation for these coefficients. What do they mean? And what are they? Are they odds-ratios? Are they marginal effects? Do they have a meaningful interpretation? Because in many cases (e.g. with probit/logit models), coefficients cannot be interpreted; only marginal effects can. In any case, I would suggest you mostly focus on Figure 1 and 3 to discuss the results of the field study: they contain all the information we need (Note: It is confusing in Table 4 to have all the control variables at the top of the table, and the coefficients we care about at the bottom.)

Thank you. To facilitate their interpretation, we now report odd ratios for coefficients of the main variables of theoretical interest for the full model 7 on pp. 15-16. Please, note that the marginal effects were already reported in the results section and their meaning was already discussed in Figures 1 and 2 in the previous version of our manuscript. They are now discussed on p. 16. We also unpack the interaction effect in Figure 3 on p. 16 by examining the adjusted predictions for social ties at representative values of project status.

We collapsed the levels of recognition (i.e., the dependent variable) into a binary variable with non-wins = 0, and recognition (honorable mention or win) = 1, and re-estimated the full model 7 with fixed effects conditional logistic regression. The results reproduce the main patterns and are presented in the robustness checks section on p. 17.

We also estimated the full model 7 with an alternative latent project quality control variable. The pattern and significance of the model results remained stable and are presented in the robustness checks section on p. 17.

The ordered logit model relies on a proportional odds assumption, i.e., that the odds are proportional across the levels of the dependent variable. In our case, the Brant test revealed deviations from the assumption of proportional odds. Leaving this aside, the estimated effects with ordered logit regression results reproduce the main patterns of the full model 7 in the paper. We report them below.

A multinomial logit model does not account for the ordinal hierarchical nature of the dependent variable and is hence not entirely appropriate for estimating the outcome of interest. Despite this critical difference, the results of multinomial logit regression reproduce the main patterns of the full model 7 in the paper. We report them below (DV level = 0 is the reference category).

Information acquisition: You often speak about how peers can use status/social-proximity to get information on the candidates. For example, on p. 5, you say “the existence of social ties between audience members and candidates should reduce the saliency of status as a signal of the quality of candidates and their work.” You also talk about this on top of p. 23. I agree that this might be an important dimension. However, I don’t think that your study can say anything about that (because, in your two experiments, jury members don’t have any private information about candidates). In a sense, your decision cuts off this channel, since by construction private information cannot play a role in your experiment. I therefore would suggest you do not speak about this with such details, but instead focus on the alternative interpretation which is that people might not like to appear to favor someone with high status to which they are socially close if the decision is public. I would, however, mention that a nice feature of your design is that it turns the “private info” channel off.

We are not entirely sure about this point. The way we have primed experiments’ participants to think about the nature of their interactions with the candidates they are expected to evaluate is closely reflective of the conditions experienced by the jurors in the Silver Tag award. In the Silver Tag award jurors must sometimes evaluate candidates with whom they have collaborated in the past, likewise participants in study 2 and 3 in the “direct ties condition” are informed that they know the commercial’s creators and collaborated with them in the past. Discriminating between the tie and non-tie condition is indeed central to our analytical effort to distinguish between the two different mechanisms that could underlie the negative interaction effect between social ties and social status. Please refer to p. 20 of the manuscript for the vignette we used to prime the prior collaboration between the evaluator and the candidate(s). Having said that, we have generally tempered our causal claims in line with your recommendation to convey a more speculative type of argumentation.

Deception. Both experiments use deception. In Study 2, you tell subjects that “the jury selects the winner collectively by disclosing the vote cast by each jury member” although this will never happen. In Study 3, it is even worse as you explicitly tell your subjects that “your vote will be publicly disclosed to the other jury members” in the public condition. I don’t think that this is something that was necessary (you could have run the experiment, either online or in the lab, without deception). Economics has a strong norm against deception, and I think this discussion applies to online studies as well. (Just like a lab loses its reputation when it deceives subjects, MTurkers know and discuss in online forums which HITs/requesters use deception.)

We appreciate your attentiveness to procedural fairness but we respectfully disagree with your claim that we deceive the participants. We are simply telling the participants how a jury evaluation process typically proceeds. First, a jury member evaluates and then reveals his/her evaluation to the other jury members for the group discussion. This is the process followed by the jury members in the Silver Tag digital competition. Our experiments aim to replicate the evaluative process of the field study, so there is no deception of the kind you seem to imply. Since juries in other award competitions (e.g., Oscars, Grammies, and so on) have a completely private process, our attempt to replicate this process in our priming is also justified. Put it differently, our priming in Study 2 and Study 3 replicates different scenarios – so deception is hardly a problem. Indeed, it would not be possible to conduct Study 2 and Study 3 without telling participants how the jury selects the winner because this information is essential for the participants to understand the evaluative setting and make their decision. In other words, telling explicitly our subjects that your vote will be (will not be) publicly disclosed to the other jury members in the public (private) condition is the necessary manipulation for conducting Study 3. The open sharing of preferences in interactive evaluative settings is common across tournament rituals as diverse as Cannes, the Venice Film Festival or the Man Booker Prize, where jurors may vouch for one or the other candidate but the adjudication of awards is always collective thus each juror’s preference is transparent to all judging panel’s members. Similar considerations hold for Study 2 in which we tell our subjects that their vote is public (i.e. it is visible to their jury peers). In sum, the two studies are not ‘deceiving’ the participants, but simply describe the evaluative setting as other experimental studies (e.g., Mueller, Melwani, Loewenstein, and Deal, 2018 – published in Academy of Management Journal; Hahl and Zuckerman, 2014 – published in American Journal of Sociology) have done as well.

Interpretation of results. I think you over-interpret a little bit your results. For example, in the conclusion you say “we demonstrated that when audience members do not have to justify their decisions before other members […]” and “social proximity tempers the effect of status on candidates’ recognition when decisions are in the public domain and, therefore, potential violations of the meritocratic ideal in social evaluation are more likely detected and stigmatized, if not punished.” In my opinion, this is quite speculative. An alternative explanation might be that people care about their social image and don’t want to be perceived as someone who favors socially-close candidates. Your conditions cannot really tease out the mechanisms why the public condition leads to these results, so I would be a little bit more modest when interpreting them. (This comment also applies to the introduction.)

Thanks for drawing our attention to this very important point. Following your suggestion, we are now more careful about the way the results should be interpreted. Accordingly, we reformulated the discussion of this finding using a more circumspect language, openly acknowledging the speculative nature of our conclusions and inviting future research to add validity to our interpretations. See in particular the revised discussion on pp. 30-31. 

p.3: It would be useful to define status at the very beginning of the text.

Status is defined as one’s relative standing in a social system in the first paragraph of the introduction.

p.3: “they are usually granted more recognition for their performance relative to low-status actors for equivalent performance”: do you have a reference for that?

Reference [8] added.

p.5: I would use “could” instead of “should” when formulating the hypotheses.

We have changed as you suggested.

p.5: I do not understand your explanation for why positive effect of status should get even stronger when high-status candidates have ties to members of the evaluating audience.

We rewrote the explanation on pp. 4-5 and articulated more thoroughly the logic. We hope the explanation is now clearer.

p.5: I don’t think the example on dyadic business relationships fits very well to your context.

We have added more background to our explanation of the substitutive effect and we are now more explicit in noting that the findings by Jensen (2003) are consistent with the logic we advance in the paper, though they are at a different level of analysis. Again, you can find these revisions on p. 5.

p.6: Why don’t you describe a bit more in the intro the context of the field study and what you find?

We added a sentence to describe the context of the field study and what we find on p. 6 of our manuscript.

p.15: Why use past tense to describe the coefficients obtained in the field study (e.g., “the coefficient for project status was 1.062”)?

You are right. We now use the present tense.

In Experiments 1+2, did your manipulation checks come before or after the measurement of your main outcome? I think in your case, there might be important demand-effects/desire for consistency.

The questions regarding our manipulation checks were always put after the measurement of the dependent variable to avoid demand effects.

I find your labeling of the different studies very confusing. Either say Field study + Online Experiment 1 and online Experiment 2, or say Study 1-2-3. But please be consistent.

We are sorry for the confusion, we now use Study 1 to refer to the Field Study, Study 2 and Study 3 to refer to the experiments reported in the main text and Study 4 and Study 5 to refer to the other experiments reported in the Appendix.

In your two vignette studies, how are treatment assigned: between-subjects, or within? I could not find this information.

For all of our experiments the assignment was between-subject. We had reported this information in the previous version of the submitted manuscript both in the “Material and Procedure” paragraph and in the “Results and Discussion” section of our experiments (please, see p. 19 and pp. 21-22 for Study 2; pp. 25-26 for Study 3; p. 48 and p. 53 for the replication experiments in the Appendix.)

Avenue for future research (p.33). This is a matter of taste, but I would not keep this paragraph.

We endeavored to streamline and prune as much as possible the avenues for research. As a result, this section is now approximately 50% shorter than in the previous version. Please note, however, that one of the research directions was added and maintained to explicitly address a comment by one of the first-round reviewers.

Again, we would like to thank you for your very thoughtful and constructive comments.

REFERENCES

Hahl O, Zuckerman EW. The denigration of heroes? How the status attainment process shapes attributions of considerateness and authenticity. American Journal of Sociology. 2014; 120(2):504-554.

Jensen M. The Role of Network Resources in Market Entry: Commercial Banks’ Entry into Investment Banking, 1991-1997. Administrative Science Quarterly. 2003; 48(3):466–97.

Mueller J, Melwani S, Loewenstein J, Deal JJ. Reframing the decision-makers’ dilemma: Towards a social context model of creative idea recognition. Academy of Management Journal. 2018; 61(1):94-110.

Reply to Reviewer #2

We very much appreciate the time and thought you have clearly put into evaluating our manuscript. Your comments have highlighted several areas where we could improve our paper. Below, we explain how we have responded to them. For the sake of convenience, we present your comments first and then our response (in bold). Please, note that we changed the title of the paper.

I commend the authors for all the work they have done to incorporate the suggestions of the esteemed fellow reviewers. I only have a couple of minor requests for clarification, mostly to improve the clarity of exposition.

Thanks so much for appreciating the work we have done so far.

- P.5 “…render neighborhoods synonymous with the activity of those who win them” could be edited for clarity.

We did it.

- P. 6: “de facto the recipients’ status boost into their own stigma,” could be edited for clarity.

We did it.

- Table 2: the mean of “Conflict of interest” is puzzling. Were about half of the evaluators in a conflict of interest? Were these people then sitting in juries they should not have sit in according to contest rules?

A juror experiences a conflict of interest if in a given contest/month a juror her/himself is involved in the project under evaluation or if the juror is employed in the same agency as one of the candidates responsible for the project. The juror experiencing a conflict of interest has to exit the jury room when that particular project is evaluated. The vote for that particular project from the juror with a conflict of interest would then be assigned as the average of the remaining jurors’ votes. The juror with a conflict of interest takes part in evaluating all the other projects in a given project month with which the juror has no conflict of interest. The awarding organization (INMA) identifies this particular case as a possible source of conflict of interest. So, we labeled the control variable accordingly. We hope this clarifies your concern.

- P. 18: “well-controlled lab study,” you use MTurk so... no lab. Also, the well-controlled can probably go. The “lab” reappears at p. 30.

Thanks bringing this inconsistency to our attention. We changed both expressions accordingly.

- I do not want to make a big fuss about the limits of vignette studies: the other reviewers already pointed those out. However, statements like “participants in the direct ties condition rated the commercial’s creators as more familiar” (p. 22) are a little awkward. Similarly, when you say that “the manipulations… were successful” (id.) I think what you mean to say is that your participants buy into what you told them in the instructions. It is an attention check.

Thank you. We revised our statements accordingly. We also understand the limits of vignettes, but recent experimental studies manipulate status and/or social ties by using scenarios similar to ours, for instance: Brands and Mehra (2019) in Academy of Management Journal, Hahl, Zuckerman and Kim (2017) in American Sociological Review, Sgourev and Althuizen (2017) in Social Psychology Quarterly, and Perry-Smith (2014) in Journal of Applied Psychology. We now cite these works to offer more anchorage to our choice.

- P. 23: “These experimental findings offer causal evidence.” I doubt you have established causation through analysis of variance. I think what you mean to say is that a different methodology corroborates the findings of your field study 1.

We agree. We no longer use ‘causal’ when referring to the mechanisms that our studies are intended to probe.

- P. 24: “disinterestedness ideal.” I would stick to the "meritocratic ideal, the term you used earlier in the paper.

Thanks for bringing this to our attention. We changed the text as you suggested.

- P. 24: “The second explanation relies on reputational arguments…”. According to this explanation, giving a prize to a buddy who is also a celebrity in the field looks bad. I would say that giving a prize to a friend will probably look bad (especially if there are quality concerns). But status and recognition in the field do not undercut reputation by a comparable magnitude to proximity. Think about the refereeing process. There is much attention to conflicts of interests that involve personal ties. I view this as a failsafe for those whose internal regulator does not work to decline refereeing requests for papers written by close friends. But no one thinks that mere prominence in a field is at odds with the “disinterestedness ideal.” Otherwise, we should prohibit every author from an R1 institution from submitting a paper (God forbid Nobel Prize winners!). That sounds a little extreme a conclusion. I would be curious to hear what the authors think. The authors make the same point at the end of page 30 and at page 6. Same page: “In the second, audience members who are socially close to the candidates whose work they are expected to evaluate are less sensitive to their status…”. I would say that if you are socially close to someone, then you should make an extra effort at evaluating the project impartially. Or perhaps, if you want to project an image of extreme resilience to social proximity considerations, you should award the prize with lower probability. I do not understand why reputation concerns command that you should disregard someone’s status when the contestant is proximate.

You are right in your interpretation. Indeed, your intuition that high social proximity could deter recognition is consistent with results reported in the study by Aadland et al. (2018). This study reflects on and documents the possibility – depending on the model specification – of an inverted U-shape relationship between nearness and recognition, suggesting that social proximity might operate as a double edged-sword if the audience–candidate relationship becomes so salient to raise suspicions of the juror’s authenticity (thus yielding concerns that inhibit rather than promote favorable evaluation on the part of the audience).

Our goal in this study is different in that we are not concerned with the effect of social proximity per se (this has already been discussed Aadland et al.’s study as well as in prior studies). Instead, we seek to explain if how and why social proximity changes the relationship between status and recognition. Not only do the results provide strong and consistent evidence across all three studies that this interaction is negative, but with study 3, we try to tease out the mechanism underlying this effect. Our interpretation is that evaluators discount the status effect at an increasing rate as they get closer to the candidates because they become overly concerned that the candidate's status would render their connection more visible, hence reproachable. In other words, the increasing proximity prompts the evaluator to "shut down" any other signal that could catalyze attention to their allocation choice. Hence, at the extreme, we should expect the incentive to reward a given candidate to be especially low when such a candidate is very close to the audience and very high in status. Our reasoning is premised on the appreciation of the role of a tie not only as a channel for the transmission of information but as a conveyer of identity, in that an actor's relations influence how others perceive the actor (White, 1981; 1992). Thus, when the tie becomes too salient due to high social proximity, this relationship deflects the award recipient's glory onto the award giver in the form of stigma. We have sought to clarify this logic in both the introduction and the conclusions (see in particular pp. 30-31). At the same time, we have attempted to recalibrate our arguments using a more circumspect and speculative language to signal that this interpretation is suggestive and that more attention to the specific cognitive and behavioral processes underlying our effect is needed to probe our understanding.

- P. 25: “These manipulations allow us to ensure that only the participants asked to evaluate (high?) status peers in the condition are (I would say might be-- again, it is a vignette) susceptible to reputation concerns.”

We changed the text as you suggested.

- P 25: just before the Methods, I would add that the other explanation, that status loses information value when you are proximate, does not lead you to predict that publicity, or lack thereof, would make a difference.

This is a very good point. We added a sentence to clarify this on p. 25 right before the method section.

- P. 25. High data quality? Do you mean that the pool of subjects is of high quality?

We no longer say ‘high data quality’ because it is unnecessary. What we wanted to say is that Prolific is a highly reputed and trusted online platform.

- P. 29: “Paramount…the roles of status and social networks…”. I believe it is better to say status and social proximity. Status is proxied as network centrality in your paper. Same on page 3.

We use ‘social network’ because we make reference to the broader literature that has looked at the impact of different types of network measures on the outcome of interest. For instance, on p. 3 and p. 28, we refer to prior social network studies in general. In both cases, therefore, it makes sense to say “Social networks …” before invoking ‘social proximity’. When we refer to what we do in our study, on the contrary, we always use ‘social proximity’.

- P. 31: The quote attributed to Dimant that “behavior that 'benefits oneself at the expense of others is socially harmful…” can go. You do not engage in welfare analysis in the paper. And the quotation reflects a pre-Coasean point of view on causation. If I consume an apple, I am benefiting myself at the expense of someone who is not eating the same apple. Dimant’s first best world, in which there are no reciprocities in resource use, is a place no one has ever inhabited.

We agree. While we still cite Dimant, we now strongly de-emphasize any discussion of welfare implications in our paper.

Again, we would like to thank you for your very thoughtful and constructive comments.

REFERENCES

Aadland, E., Cattani, G., S. Ferriani (2018) “The Social Structure of Consecration in Cultural Fields: The Influence of Status and Social Distance in Audience-Candidate Evaluative Processes”, In C. Jones M. Maoret (eds.), Frontiers of Creative Industries: Exploring Structural and Categorical Dynamics. Research in the Sociology of Organizations, 2018, 55: 129–157.

Brands RA, Mehra A. Gender, brokerage, and performance: a construal approach. Academy of Management Journal. 2019; 62(1):196-219.

Hahl O, Zuckerman EW, Kim M. Why elites love authentic lowbrow culture: Overcoming high-status denigration with outsider art. American Sociological Review. 2017; 82(4):828-856.

Perry-Smith, J. E. Social network ties beyond nonredundancy: An experimental investigation of the effect of knowledge content and tie strength on creativity. Journal of applied psychology. 2014;99(5), 831.

Sgourev SV, Althuizen N. Is it a masterpiece? Social construction and objective constraint in the evaluation of excellence. Social Psychology Quarterly. 2017; 80(4):289-309.

 

Reply to Reviewer #3

We very much appreciate the time and thought you have clearly put into evaluating our manuscript. Your comments have highlighted several areas where we could improve our paper. Below, we explain how we have responded to them. For the sake of convenience, we present your comments first and then our response (in bold). Please, note that we changed the title of the paper.

I don’t know this precise field of literature well, so I would leave it to another referee to judge the novelty and originality of the contribution this study makes to the existing literature. If the empirical/experimental evidence is sufficiently novel, I think the topic is very relevant and interesting in many areas and professions. One application certainly is peer review in academia.

Thanks for appreciating our paper.

STUDY DESIGN:

• It’s a nice feature of the article, that the authors try to establish additional external validity and try to validate their own interpretation of their findings by interviewing professionals in the field.

• FIELD STUDY: Can the authors elaborate a bit more on how project quality is controlled for? It seems like this is not perfectly captured by their variables and it is very likely to be correlated to project status. This might be a problem for the identification in the field setting.

We control for project quality in our field study by incorporating variables that jointly represent a proxy for project quality. In particular, our interviews with field insiders revealed that high quality projects tend to exhibit certain observable and measurable characteristics that are strongly correlated with quality. First, high-quality projects tend to be technologically advanced and innovative in terms of technological application. With the diffusion of broadband technology and increasing downloading speeds, digital advertising professionals have now the opportunity to create more sophisticated creative solutions with visually appealing interactive content based on video/film, sound, 3D animation and streaming technologies. Second, high-quality projects make use of ample resources in terms of budget size and work hours in order to develop more ambitious solutions. Larger projects, proxied by the number of project participants, also increase the likelihood of social ties between jury members and members of the project team. Although other unobserved characteristics might affect project quality, the technical sophistication of a project and the number of people working on it represent a reasonably good approximation of a project’s underlying quality. In the robustness check section, we further explain how we endeavored to control for project quality using a different approach.

We also estimated the full model 7 with an alternative latent project quality control variable. The pattern and significance of the model results remained stable and are presented in the robustness checks section on p. 17.

• EXPERIMENT 1:

o I find it a bit problematic that the vignettes only contain information about creators social status and social ties (p.20). This makes the situation highly hypothetical and also can create experimenter demand effects.

o The social ties condition is highly abstract--“imagine you have collaborated with them in the past”—and does not really induce true social ties. There are many experiments that truly manipulate “social proximity”, i.e. that really induce a social distance or connection but priming them on common preferences, creating groups. It would have been even possible here to actually have subjects work together on a task and then match them for the jury-candidate experiment… There are many good examples in previous studies and I don’t understand why the authors don’t use them. I’m not convinced this manipulation has the desired effect.

We understand your concerns of using vignettes. As explained in footnote 9, we relied on prior similar experimental studies that used descriptive texts to prime status or social ties, in particular: Brands and Mehra (2019) in Academy of Management Journal, Hahl, Zuckerman and Kim (2017) in American Sociological Review, Sgourev and Althuizen (2017) in Social Psychology Quarterly, and Perry-Smith (2014) in Journal of Applied Psychology.

We also thought about designing and running a lab experiment to create groups with in-person subjects. However, it was not possible to do it in a reasonable time for resubmitting our manuscript. Unfortunately, running lab experiments is currently restricted due to the COVID-2019 situation – and most likely it will be restricted till the end of this year.

We thus decided to run a new study, reported in the Appendix as Study 5, in which where we accompanied the descriptive scenario with a figure representing the collaborative network of the evaluators and their ties to the commercial’s creators. This Study 5 confirms that status reduces the probability of rewarding socially proximate candidates.

o DV (p.21): Subjects watch one commercial and should then say how likely they are to award it. How should they make this choice? They have nothing to base their choice on, i.e. no other commercials to compare it to. Because of this, the manipulations by the experimenter might actually have a larger effect than they would have otherwise.

Again, we understand your concerns. This was a design choice based on the consideration that we all watch commercials and therefore have some terms of comparison. There are also many exemplary studies in organizational scholarship that ask participants to express their evaluations based on a single product, idea, or funding campaign (e.g., Clarke, Cornelissen and Healey, 2019 – published in Academy of Management Journal; Mueller, Melwani, Loewenstein, and Deal, 2018 – published in Academy of Management Journal; Greenberg and Mollick, 2016 – published in Administrative Science Quarterly).

• EXPERIMENT 2:

o The dampened signaling effect of status if a jury member and a candidate have close social ties does NOT work in their design because they HAVE NO actual social ties. This mechanism cannot be detected in their experiment (p.24). This also shows that their manipulation of social proximity is highly problematic for their purpose (see above).

Frankly, we do not fully understand your comment. In experiment 2 (i.e., Study 3), we primed all participants by telling them that they are expected to evaluate candidates with whom they have a tie (i.e., they worked with them in the past). In other words, all of them evaluated the commercial under the ‘social (direct) ties” condition. If you meant to say that our experimental design cannot capture the existence of a tie, please see our response to your previous comment in which we mention a few noticeable studies that have followed our approach to manipulate the existence of ties. Please, also refer to our additional study 5 where we follow an alternative approach to priming ties. We also emphasize that in the presence of ties between the evaluators and the candidates, status has a dampening effect only when the evaluation is public, but no effect when the evaluation is private (non-public) – as we explain on pp. 28-31 of the paper.

ADDITIONAL MINOR COMMENTS

• In Experiment 1, the authors mix “status” and “expertise” in their treatment manipulation which I find very questionable. What is the reason for doing this? It seems easy to disentangle the two and therefore to avoid this confound. Giving an “expert cue” and interpreting its effect as “status” seems like an interpretation that is probably not warranted.

In Study 2 and Study 3, we operationalized status as “well-known experts” because prior sociological research in the creative industry often does not conceptually discern between status and expertise as both are considered a signaling strategy (e.g., Jones, 2002). For instance, in the film industry, Elsbach and Kramer (2003) argue that when catchers were “aware of their greater expertise and ability during a pitch, they were likely to categorize their relationship with the pitcher as one between a high-status expert and a low-status incompetent” (p. 294). Thus, this suggests that status and expertise are not necessarily considered conceptually distinct in the creative industry. Similar considerations pertain to sociological works that use scales including prestigious, competent and skilled to measure status (Hahl, Oliver, and Zuckerman, 2014; Hahl, Zuckerman and Kim, 2017)

Furthermore, in our status manipulation we qualified ‘experts’ as ‘well-known’ to assign prestige to the commercial’s creators. Indeed, our manipulation checks for status indicated that these commercial’s creators were perceived to be more prestigious advertisers than advertisers described as ‘non-experts.’

Finally, to address concerns of a reviewer that were similar to yours, in the first round of our revision we replicated Study 2 with a different manipulation of status in which we used ‘famous’ and ‘not very famous’ to describe the commercial’s creators. Specifically, we mentioned in the status condition that ‘the authors of the commercial are famous professionals in advertising;’ while, in the no-status condition we specified that ‘the authors of the commercial are not very famous professionals in advertising.’ Our manipulation check revealed the effectiveness of this status manipulation; also, our results replicated the findings obtained in Study 2. Given the prior literature pointing to the importance of social ties in shaping a producer’s fame (Lang and Lang 1988; Accominotti 2009), this approach should temper your preoccupations with the potential disconnect in status operationalization criteria between Study 1 (centrality) and Study 2 and Study 3 (expertise).

Please, refer to the Study 4 in the Appendix for a complete description of this additional experiment.

• Especially without pre-registration, removing any subjects from the analysis is questionable, e.g. removing the outliers, p.22.

We used the 2.5 SD cut off point as recommend by scholars (Van Selst M and Jolicoeur, 1994; Meyvis and Van Osselaer, 2017) and applied it consistently across all experimental studies. We did the same for the other criteria we used to detect careless responses.

• Overall, I recommend to shorten and streamline the text.

We did as you suggested.

• Tables in the text instead of in the appendix would have helped a lot to make the paper more readable.

The tables in the Appendix refer exclusively to additional studies that we conducted as robustness checks. We now provide a summary description of these studies in the main body of the paper but it would be confusing to include these tables in the text. They belong to the Appendix. All tables/figures that refer to the experiments described in the paper are before the Appendix.

Again, we would like to thank you for your very thoughtful and constructive comments.

REFERENCES

Accominotti F. Creativity from interaction: Artistic movements and the creativity careers of modern painters. Poetics. 2009; 37(3), 267-294.

Brands RA, Mehra A. Gender, brokerage, and performance: a construal approach. Academy of Management Journal. 2019; 62(1): 196-219.

Clarke JS., Cornelissen JP, Healey MP. Actions speak louder than words: How figurative language and gesturing in entrepreneurial pitches influences investment judgments. Academy of Management Journal. 2019; 62(2): 335-360.

Elsbach KD, Kramer RM. Assessing creativity in Hollywood pitch meetings: Evidence for a dual-process model of creativity judgments. Academy of Management journal. 2003; 46(3): 283-301.

Greenberg J, Mollick E. Activist choice homophily and the crowdfunding of female founders. Administrative Science Quarterly. 2017; 2(2): 341-374.

Hahl O, Zuckerman EW, Kim M. Why elites love authentic lowbrow culture: Overcoming high-status denigration with outsider art. American Sociological Review. 2017; 82(4): 828-856.

Hahl O, Zuckerman EW. The denigration of heroes? How the status attainment process shapes attributions of considerateness and authenticity. American Journal of Sociology. 2014; 120(2): 504-554.

Jones C. Signaling expertise: How signals shape careers in creative industries. Career creativity: Explorations in the remaking of work. 2002; 209-228.

Lang GE, Lang K. Recognition and Renown: The Survival of Artistic Reputation. American Journal of Sociology. 1988; 94 (1): 79-109.

Meyvis T, Van Osselaer SMJ. Increasing the Power of Your Study by Increasing the Effect Size. Journal of Consumer Research. 2017; 44(5):1157-173.

Mueller J, Melwani S, Loewenstein J, Deal JJ. Reframing the decision-makers’ dilemma: Towards a social context model of creative idea recognition. Academy of Management Journal. 2018; 61(1): 94-110.

Perry-Smith, J. E. Social network ties beyond nonredundancy: An experimental investigation of the effect of knowledge content and tie strength on creativity. Journal of applied psychology. 2014; 99(5), 831.

Sgourev SV, Althuizen N. Is it a masterpiece? Social construction and objective constraint in the evaluation of excellence. Social Psychology Quarterly. 2017; 80(4):289-309.

Van Selst M, Jolicoeur P. A Solution to the Effect of Sample Size on Outlier Elimination. The Quarterly Journal of Experimental Psychology. 1994; 47(3):631-50.

---

## [Decision Letter · Decision Letter 1]

13 Aug 2020

PONE-D-20-09850R1

"Reflecting Glory or Deflecting Stigma? The Interplay between Status and Social Proximity in Peer Evaluations"

PLOS ONE

Dear Dr. cattani,

Thank you for submitting your manuscript to PLOS ONE. After careful consideration, we feel that it has merit but does not fully meet PLOS ONE’s publication criteria as it currently stands. Therefore, we invite you to submit a revised version of the manuscript that addresses the points raised during the review process  

We look forward to receiving your revised manuscript.

Kind regards,

Valerio Capraro

Academic Editor

PLOS ONE

Additional Editor Comments (if provided):

One of the reviewers still suggests some improvements.

Reviewers' comments:

Reviewer's Responses to Questions

**Comments to the Author**

1. If the authors have adequately addressed your comments raised in a previous round of review and you feel that this manuscript is now acceptable for publication, you may indicate that here to bypass the “Comments to the Author” section, enter your conflict of interest statement in the “Confidential to Editor” section, and submit your "Accept" recommendation.

Reviewer #1: All comments have been addressed

Reviewer #2: All comments have been addressed

Reviewer #3: (No Response)

2. Is the manuscript technically sound, and do the data support the conclusions?

Reviewer #1: Yes

Reviewer #2: Yes

Reviewer #3: Partly

3. Has the statistical analysis been performed appropriately and rigorously? 

Reviewer #1: Yes

Reviewer #2: Yes

Reviewer #3: Yes

4. Have the authors made all data underlying the findings in their manuscript fully available?

Reviewer #1: Yes

Reviewer #2: No

Reviewer #3: Yes

5. Is the manuscript presented in an intelligible fashion and written in standard English?

Reviewer #1: Yes

Reviewer #2: Yes

Reviewer #3: Yes

6. Review Comments to the Author

Reviewer #1: The authors have addressed my comments in a satisfactory way and I am happy to accept the paper in its current version (although I think it is still a bit too long and could be streamlined substantially; which is a comment I already made in my first assessment of the paper).

Reviewer #2: (No Response)

Reviewer #3: The authors re-write part of the paper and provided additional evidence to address some of the reviewers’ concerns. Overall, I appreciate that the authors provide new evidence and modified some parts in the paper, however, the changes do not fully address some major concerns I had (have) with the paper. Therefore, I was on the fence whether to reject or accept the paper before and the revised version has not convinced me to change my mind in either direction.

Some comments on the revised version:

Study 5

Study 5 replicates part of study 2 by varying status when social ties are present and evaluations are public. By modifying the way close social ties are depicted in the experiment, the authors provide a robustness effect of the finding that with close social ties, higher status negatively affects the likelihood to give an award. Contrary to study 2, social ties are not varied and the interaction effect is not tested. While providing a robustness check is valuable per se, I have two questions: Which exact concern does this design variant address? Why is the interaction not tested, i.e. why is only part of the result from study 2 replicated? I suggest to either address these two questions and explain the value added or to just leave out this replication in the final paper.

Study 3

The authors suggest that two mechanisms can drive an interaction of status and social ties: information and social pressure. Study 3 varies whether evaluations are public or not. Social ties are kept constant (close ties) and status is varied as before.

p. 28: “…we could rule out the alternative explanation that status and social (ties) information operate as substitutive information devices” As also pointed out in my first round review, I think the authors have to be more careful with their conclusion here. (1) I agree that it’s probably unlikely to play a major role in your experiment due to the other previous findings, but the conclusion is too strong here, it cannot be “ruled out”. (2) information is a mechanism that they cannot test with the current design since there is no information that comes with close social ties in the way they are modelled in the experiment (purely based on descriptions in the vignettes). Thus, this channel seems to be shut down in this study, i.e. they don’t test it (which is okay, just don’t state otherwise).

7. PLOS authors have the option to publish the peer review history of their article (what does this mean?). If published, this will include your full peer review and any attached files.

Reviewer #1: No

Reviewer #2: No

Reviewer #3: No

---

## [Author Response · Author response to Decision Letter 1]

19 Aug 2020

Reply to Editor

We appreciate the time and thought you have put into your evaluation of our manuscript. Since the first two reviewers accepted the paper, we only addressed the comments of reviewer 3.

We hope you will find this revised version of our manuscript to deserve publication in PLOS ONE. Again, thanks for your guidance throughout the review process. We remain at your disposal for any further clarification.

 

Reviewer #1: Thanks for accepting the paper.

Reviewer #2: Thanks for accepting the paper.

Reply to Reviewer #3: Thanks for your comments. Below, we respond to them in bold.

The authors re-write part of the paper and provided additional evidence to address some of the reviewers’ concerns. Overall, I appreciate that the authors provide new evidence and modified some parts in the paper, however, the changes do not fully address some major concerns I had (have) with the paper. Therefore, I was on the fence whether to reject or accept the paper before and the revised version has not convinced me to change my mind in either direction.

Again, we very much appreciate the time and thought you have put into re-evaluating our manuscript.

Study 5

Study 5 replicates part of study 2 by varying status when social ties are present and evaluations are public. By modifying the way close social ties are depicted in the experiment, the authors provide a robustness effect of the finding that with close social ties, higher status negatively affects the likelihood to give an award. Contrary to study 2, social ties are not varied and the interaction effect is not tested. While providing a robustness check is valuable per se, I have two questions: Which exact concern does this design variant address? Why is the interaction not tested, i.e. why is only part of the result from study 2 replicated? I suggest to either address these two questions and explain the value added or to just leave out this replication in the final paper.

We designed Study 5 to focus on the case with social ties because we thought this case was theoretically more interesting. This was also in line with Study 3, where we kept social ties constant to disentangle the mechanism underlying the negative effect of status. Furthermore, we had already showed the existence of the interaction between status and social ties in Study 1 (Field Study), Study 2 (experiment) and Study 4 (experiment). Following your suggestion, we decided to remove Study 5 from the revised version of our manuscript: this study was indeed only a robustness check and it is no longer necessary in the final version.

Study 3

The authors suggest that two mechanisms can drive an interaction of status and social ties: information and social pressure. Study 3 varies whether evaluations are public or not. Social ties are kept constant (close ties) and status is varied as before.

p. 28: “…we could rule out the alternative explanation that status and social (ties) information operate as substitutive information devices” As also pointed out in my first round review, I think the authors have to be more careful with their conclusion here. (1) I agree that it’s probably unlikely to play a major role in your experiment due to the other previous findings, but the conclusion is too strong here, it cannot be “ruled out”. (2) information is a mechanism that they cannot test with the current design since there is no information that comes with close social ties in the way they are modelled in the experiment (purely based on descriptions in the vignettes). Thus, this channel seems to be shut down in this study, i.e. they don’t test it (which is okay, just don’t state otherwise).

You make a very important point. Following your suggestion, we toned down our conclusion.

We hope we have addressed all of your comments. Again, thanks for your very thoughtful and constructive comments.

---

## [Editor Report · Decision Letter 2]

21 Aug 2020

"Reflecting Glory or Deflecting Stigma? The Interplay between Status and Social Proximity in Peer Evaluations"

PONE-D-20-09850R2

Dear Dr. cattani,

We’re pleased to inform you that your manuscript has been judged scientifically suitable for publication and will be formally accepted for publication once it meets all outstanding technical requirements.

Kind regards,

Valerio Capraro

Academic Editor

PLOS ONE
---

## [Editor Report · Acceptance letter]

15 Sep 2020

PONE-D-20-09850R2

Reflecting Glory or Deflecting Stigma?
The Interplay between Status and Social Proximity in Peer Evaluations

Dear Dr. Cattani:

I'm pleased to inform you that your manuscript has been deemed suitable for publication in PLOS ONE. Congratulations! Your manuscript is now with our production department.

Kind regards,

on behalf of

Dr. Valerio Capraro 

Academic Editor

PLOS ONE